# PROPAGATING DISTRIBUTIONS THROUGH NEURAL NETWORKS

## ABSTRACT

We propose a new approach to propagating probability distributions through neural networks. To handle non-linearities, we use local linearization and show this to be an optimal approximation in terms of total variation for ReLUs. We demonstrate the advantages of our method over the moment matching approach popularized in prior works. In addition, we formulate new loss functions for training neural networks based on distributions. To demonstrate the utility of propagating distributions, we apply it to quantifying prediction uncertainties. In regression tasks we obtain calibrated confidence intervals, and in a classification setting we improve selective prediction on out-of-distribution data. We also show empirically that training with our uncertainty aware losses improve robustness to random and adversarial noise.

## 1 INTRODUCTION

Neural networks are routinely used in applications affecting our daily lives, including safety-critical domains. As a result, quantifying uncertainties in neural network decisions and improving their robustness against noise has become an important problem. A prominent example is autonomous driving [1], where a neural network is not only supposed to detect and classify various objects like other cars or pedestrians on the road, but also to know how certain it is about this decision and to allow, e.g., for human assistance for uncertain cases. As pointed out by many works, e.g., by Kendall and Gal [2] or by Kiureghian *et al.* [3], prediction uncertainties can arise from two different sources: systematic uncertainty in the data, which is referred to as epistemic uncertainty, or random uncertainties in the data, e.g., because of noisy sensors, which is referred to as aleatoric uncertainty [4]. Both types of uncertainty, epistemic and aleatoric, have received considerable attention. Works focusing on epistemic uncertainties include out-of-distribution (OOD) detection [5], e.g., via orthogonal certificates [6] or via Bayesian neural networks [7–10]. Other works focus on aleatoric uncertainties, e.g., via uncertainty propagation [1, 11–13], via ensembles [14], via simultaneous quantile regression [6], or also via Bayesian neural networks [15]. Note that many of the aforementioned methods can be applied to quantify both types of uncertainties. Similarly, in this work, we focus on both aleatoric and epistemic uncertainties.

We consider the problem of evaluating $f(x + \epsilon)$, where $f$ is a neural network, $x$ is an input data point, and $\epsilon$ is a random noise variable. Therefore, $f(x + \epsilon)$ is a random variable, where its mean is used for prediction, and its variance for quantifying the uncertainty. This perspective allows assessing the sensitivity of the neural network for uncertainty quantification. Aleatoric uncertainty due to input measurement errors can be estimated by modeling how much a prediction changes under a respective input uncertainty. Incorporating the sensitivity of a neural network during training can also allow epistemic uncertainty quantification: when the output variance is regularized (as in our proposed loss function), the variance of predictions for observed data is minimized. Thus, for the regions where data does not provide sufficient information, the variance can be larger. We visualize the uncertainty quantification on a toy problem in Fig. 1: on the left our method identifies high uncertainty in the regions where samples from two classes overlap corresponding to aleatoric uncertainty; on the right, we capture epistemic uncertainty due to many possible decision boundaries separating the two classes. See figure 6 in Hüllermeier *et al.* [16] for an analogous example with linear models.

Due to the high complexity of neural networks, specifically many non-linear functions, it is intractable to compute the distribution of $f(x + \epsilon)$ analytically. A straightforward way to approximate the mean and (co-)variances of $f(x+\epsilon)$ is to use the Monte Carlo method. Unfortunately, this is not suitable for

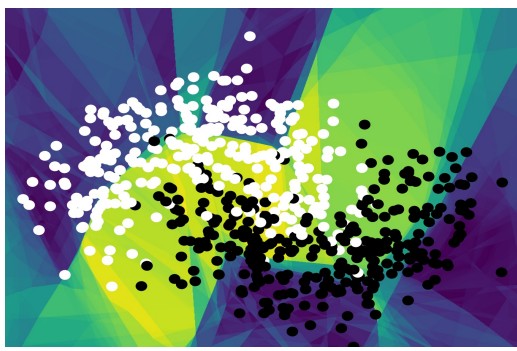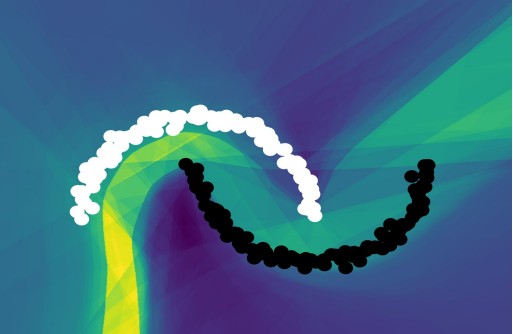

Figure 1: Aleatoric (left) and epistemic (right) uncertainty estimation. Yellow indicates a large estimated uncertainty and blue indicates a small estimated uncertainty.

the problems we consider: evaluating a neural network $f$ many times is computationally prohibitive, and the quality of the approximation for the (co-)variances as well as the quality of the gradients deteriorates quickly when the data is high-dimensional, e.g., images. Instead, we consider analytical parametric approximations of $f(x + \epsilon)$. The key challenge is to approximate the transformations of distributions by non-linearities such as ReLU. The widely used moment matching technique [17] computes mean and variance of the corresponding transformed distribution and uses a Gaussian distribution with the same mean and variance as the approximation. Moment matching can also be applied for other distributions of the exponential distribution family [18]. However, this procedure requires that the moments are defined and finite, which is not always the case, e.g., for Cauchy distributions. Also, it cannot be applied to conventionally pre-trained deep neural networks. In addition, in our empirical studies, we observed that training with moment matching is numerically unstable, requiring careful hyperparameter tuning to obtain meaningful results.

**Contributions**

• We propose an approximation of a distribution transformed by a ReLU non-linearity that minimizes the total variation distance to the true distribution. Empirically, we show that this approximation is also effective in approximating transformations with other popular activation functions, and applicable to multivariate and multilayer cases.

• We propose a new loss function for learning with Gaussian and Cauchy input distributions.

• We show that our method can quantify aleatoric uncertainty by obtaining calibrated confidence intervals in regression and epistemic uncertainty by improving selective prediction in classification.

• We demonstrate that our method improves robustness against random and adversarial noise.

Our PyTorch-based [19] framework for propagating distributions will be publicly available.

## 2 RELATED WORK

Several approaches have been suggested to introduce some treatment of uncertainty into neural networks. They can be grouped into three broad categories, although there are also hybrid approaches.

The first category tries to model uncertainty via sampling. Here, we find, among others: Variational inference for neural networks [9], a tractable approximation to Bayesian inference for neural networks; Bayes by Backprop [8], which learns a probability distribution by Monte Carlo sampling to introduce uncertainty in the weights of the network; Monte Carlo dropout [20], which estimates models' prediction uncertainties by applying dropout [21] at test time. An overview of current techniques for Bayesian deep learning in the context of computer vision models can be found in Gustafsson *et al.* [22]. For aleatoric uncertainties Bouchacourt *et al.* [23] minimize the dissimilarity coefficient between the true and an estimated distribution (modeled by a neural network). The drawback of these methods is the computationally expensive inference, i.e., to make a single prediction they require many forward passes of a neural network.

The second category of approaches concentrates on modeling uncertainties without sampling, i.e., analytically. These works approximately propagate normal distributions at individual training samples through the network to model the network response to perturbed inputs [11, 12, 18, 24]. Wang *et al.* [18] propose Natural-Parameter networks, which allow using exponential-family distributions to model weights and neurons. Similarly, Wang *et al.* [25] and Postels *et al.* [26] use uncertainty propagation to sampling-free approximate dropout. These methods use moment matching for propagating parametric probability distributions through neural networks, as we will discuss in greater detail subsequently. Wu *et al.* [27] propose an approximation to moment matching of Gaussian distributions with covariance for ReLU non-linearities to provide a deterministic variational inference approximation (DVIA) for Bayesian neural networks.

The third category of approaches estimates uncertainties by training a neural network to predict not only an output value but also the output value's uncertainty. Lakshminarayanan *et al.* [14] use neural networks for predicting Gaussian distributions instead of individual values and extend this idea with deep ensembles. Tagasovska *et al.* [6] learn conditional quantiles for aleatoric uncertainty estimation and propose orthonormal certificates for quantifying epistemic uncertainty. We use several methods from this category as baselines in our experiments.

Our approach falls into the *second* category and is most similar to Gast *et al.* [11] and Shekhovtsov *et al.* [12]. Our approach differs in how distributions are passed through non-linearities such as ReLUs. Previous work relies on moment matching [17], also known as assumed density filtering. Moment matching is a technique where the first two moments of a distribution, such as the output distribution of ReLU (i.e., the mean and variance) are computed and used as parameters for a normal distribution to approximate the true distribution. As it is otherwise usually intractable, moment matching assumes diagonal covariances. In our approach, we propagate normal distributions by an approximation that minimizes the total variation (TV) distance for ReLU activations. We find that our approximation is faster to compute, a better approximation with respect to TV, and allows propagating full covariances. Further, as we do not rely on moment matching, we can also propagate Cauchy distributions for which the moments are not finitely defined. Another advantage of our method is that it can be applied to pre-trained models. On the other hand, moment matching changes the mean such that there are deviations from what a network without uncertainty propagation would compute, leading to poor predictions when applied to a pre-trained network. Lastly, instead of approximating the distribution of the output classes for the loss function [11, 12], we propose a new loss function suitable for distributional outputs.

## 3 Propagating Distributions through Neural Networks

For propagating parametric distributions through neural networks, we consider affine transformations and non-linearities, such as ReLU, separately. For affine transformations, exact computation of the parametric output distribution is possible due to the reproductive property of Gaussian and Cauchy distributions. For non-linearities, we use local linearization and show that this approximation minimizes the TV distance to the true intractable distribution for ReLUs. Recall that in our setting, the input to a neural network is a random variable and the weights are learned parameters.

### 3.1 Affine Transformations

Fully connected layers as well as convolutional layers are affine transformations of their inputs.

For fully connected layers, we use the notation $\boldsymbol{y} = \boldsymbol{x}\boldsymbol{A}^\top + \boldsymbol{b}$ where $\boldsymbol{A} \in \mathbb{R}^{m \times n}$ is the weight matrix and $\boldsymbol{b} \in \mathbb{R}^{1 \times m}$ is the bias vector with $n, m \in \mathbb{N}_+$.

As convolutional layers can be expressed as fully connected layers, we discuss them in greater detail in Supplementary Material B and only discuss fully connected layers here.

Given a *multivariate normal distribution* $\boldsymbol{X} \sim \mathcal{N}(\boldsymbol{\mu}, \boldsymbol{\Sigma})$, $\boldsymbol{\mu} \in \mathbb{R}^{1 \times n}, \boldsymbol{\Sigma} \in \mathbb{R}^{n \times n}$, $\boldsymbol{X}$ can be transformed by a fully connected layer via $\boldsymbol{\mu} \mapsto \boldsymbol{\mu}\boldsymbol{A}^\top + \boldsymbol{b}$ and $\boldsymbol{\Sigma} \mapsto \boldsymbol{A}\boldsymbol{\Sigma}\boldsymbol{A}^\top$.

Given a *multivariate normal distribution without covariances* $\boldsymbol{X} \sim \mathcal{N}(\boldsymbol{\mu}, \boldsymbol{\sigma}^2), \boldsymbol{\mu} \in \mathbb{R}^{1 \times n}, \boldsymbol{\sigma} \in \mathbb{R}^{1 \times n}$, $\boldsymbol{X}$ can be transformed by a fully connected layer as follows: $\boldsymbol{\mu} \mapsto \boldsymbol{\mu}\boldsymbol{A}^\top + \boldsymbol{b}$ and $\boldsymbol{\sigma}^2 \mapsto \boldsymbol{\sigma}^2(\boldsymbol{A}^2)^\top$ where $\cdot^2$ denotes the element-wise square.

Given a *multivariate Cauchy distribution* $X \sim \mathcal{C}(x_0, \gamma)$, $x_0 \in \mathbb{R}^{1 \times n}, \gamma \in \mathbb{R}^{1 \times n}$, $X$ can be transformed by a fully connected layer by $x_0 \mapsto x_0 A^\top + b$ and $\gamma \mapsto \gamma \operatorname{Abs}(A)^\top$.

Notably, in all cases, the location $\mu/x_0$ does coincide with the values propagated in conventional network layers. This allows applying this method directly to conventionally trained neural networks.

*Average pooling* down-samples by averaging the values of pooling regions. As this is a linear combination, it can be expressed using matrix multiplications and thus needs no special treatment.

## 3.2 NON-LINEAR TRANSFORMATIONS

To handle non-linearities, we utilize local linearization. That is, we transform the mean / median and the variance / scale as follows:

$$(\mu, \sigma) \mapsto (f(\mu), f'(\mu) \cdot \sigma) \tag{1}$$

for univariate distributions and as

$$(\boldsymbol{\mu}, \boldsymbol{\Sigma}) \mapsto (f(\boldsymbol{\mu}), f'(\boldsymbol{\mu})\boldsymbol{\Sigma} f'(\boldsymbol{\mu})^\top) \tag{2}$$

for multivariate distributions. ReLU is the most common non-linearity for neural networks, and we now study it in more detail. Following local linearization (Eq. 1), our approximation for transforming distributions with ReLUs is

$$\operatorname{ReLU}: (\mu, \sigma) \mapsto \begin{cases} (\mu, \sigma) & \mu \geq 0 \\ (0, 0) & \text{otherwise} \end{cases} \tag{3}$$

for distributions parameterized via $\mu$ and $\sigma$. In fact, for Gaussian and Cauchy distributions, this approximation is optimal wrt. TV. The TV between two probability distributions, i.e., between the true distribution $Q$ and an approximation $P$, is defined as

$$TV(\boldsymbol{P}, \boldsymbol{Q}) = \sup_A \left| \int_A (p - q) \, d\nu \right| = \frac{1}{2} \int |p - q| \, d\nu. \tag{4}$$

To motivate our choice of the approximation quality metric, we note that TV is a proper distance metric on probability distributions. Specifically, a TV of 0 implies that two distributions are the same, i.e., all their moments (including means) are equal. Devroye *et al.* [28] show that TV upper bounds the maximum of differences between moments for Gaussian distributions.

In the following theorem, we formalize that our approximation of the ReLU non-linearity minimizes the TV. Note that when parameterizing a Gaussian or Cauchy distribution with $(0, 0)$, we are referring to Dirac's $\delta$ distribution. An illustration of the transformed distributions and their approximations can be found in Supplementary Material A.

**Theorem 1.** *Local linearization provides the best Gaussian approximation of a Gaussian distribution transformed by a ReLU non-linearity with respect to the total variation:*

$$\operatorname*{arg\,min}_{(\tilde{\mu}, \tilde{\sigma})} TV(\boldsymbol{P}, \boldsymbol{Q}) = \begin{cases} (\mu, \sigma) & \mu \geq 0 \\ (0, 0) & \text{otherwise} \end{cases} \tag{5}$$

$$\text{where } \boldsymbol{P} = \mathcal{N}(\tilde{\mu}, \tilde{\sigma}^2), \quad \boldsymbol{Q} = \operatorname{ReLU}(\mathcal{N}(\mu, \sigma^2))$$

*Proof.* We distinguish 3 cases, $\mu < 0$, $\mu = 0$, and $\mu > 0$:

($\mu < 0$)   In the first case, the true distribution has a probability mass of $p_0 > 0.5$ at 0 because all values below 0 will be mapped to 0 and $\operatorname{CDF}(0) > 0.5$. As the approximation is parameterized as $(0, 0)$, it has a probability mass of 1 at 0. Therefore, the TV is $|1 - p_0| < 0.5$. All parameterized distributions where $\sigma > 0$ have no probability mass at 0 and thus a TV of at least $|0 - p_0| = p_0 > 0.5$.

($\mu = 0$)   In this case, $\int_0^\infty |p - q| \, d\nu = 0$ because the distributions are equal on this domain. The true distribution has a probability mass of $p_0 = 0.5$ at 0, therefore the TV is $|0 - p_0| = 0.5$. In fact, all distributions with $\sigma > 0$ have a TV of at least 0.5 because $|0 - p_0| = 0.5$. The distribution parameterized by $(0, 0)$ has the same TV at this point.

($\mu > 0$)   In this case, $p_0 = \operatorname{CDF}(0) < 0.5$. Further, again $\int_0^\infty |p - q| \, d\nu = 0$. Thus, the TV of our distribution is $|0 - p_0| = p_0 < 0.5$. All distributions with $\sigma > 0$ have a TV of at least $p_0$ because $|0 - p_0| = p_0$. The distribution parameterized by $(0, 0)$ has a TV of $|1 - p_0| > 0.5$.

Thus, the approximation by linearization (Eq. 3) is the optimal approximation wrt. total variation.   $\square$

**Corollary 1.** *Theorem 1 also applies to Cauchy distributions parameterized via $x_0, \gamma$ instead of $\mu, \sigma$.*

*Proof.* The proof of Theorem 1 also applies here. $\square$

To demonstrate the accuracy of this approximation in the multi-dimensional and multi-layer case, we simulate multi-layer neural networks with ReLU activations and evaluate the TV between parametric approximations and Monte Carlo approximation with

Table 1: Simulation of propagating normal distributions through a neural net. Reported is the intersection of probability mass $(1 - TV)$.

| $\sigma$ | With Cov. (Ours) | DVIA [27] | Without Cov. (Ours) | Moment Matching |
|---|---|---|---|---|
| 0.1 | **0.9791** $\pm$ 0.0202 | 0.9720 $\pm$ 0.0228 | 0.2361 $\pm$ 0.0250 | 0.2611 $\pm$ 0.0256 |
| 1 | **0.8747** $\pm$ 0.0546 | 0.8519 $\pm$ 0.0543 | 0.2243 $\pm$ 0.0237 | 0.2195 $\pm$ 0.0294 |
| 10 | **0.7586** $\pm$ 0.0407 | 0.7035 $\pm$ 0.0366 | 0.2186 $\pm$ 0.0244 | 0.1976 $\pm$ 0.0179 |
| 100 | **0.6877** $\pm$ 0.0333 | 0.5845 $\pm$ 0.0479 | 0.2261 $\pm$ 0.0282 | 0.1724 $\pm$ 0.0111 |
| 1000 | **0.6808** $\pm$ 0.0318 | 0.5318 $\pm$ 0.0516 | 0.2193 $\pm$ 0.0248 | 0.1706 $\pm$ 0.0109 |

$10^6$ samples representing the oracle. We use a neural network with 4 hidden layers and 100 neurons per layer with ReLU non-linearities trained with softmax cross-entropy on the Iris data set. This data set has small input and output dimensionalities, allowing us to use Monte Carlo as a feasible and accurate oracle estimator of the truth. The results of this simulation are displayed in Tab. 1.

The simulation shows that propagating the covariances produces the best estimates of the output distribution. Further, the simulation shows that when discarding covariances after each layer, on average, the approximation from Eq. 3 performs better than moment matching. Note that moment matching with covariances is intractable. Therefore, we use the deterministic variational inference approximation method (DVIA) by Wu *et al.* [27] in this case as a baseline. As DVIA has a significantly larger computational complexity (its computational overhead is linear in the size of the largest layer; see Section 5.5 for additional details), we can compare our method only in these simulation experiments to DVIA. Overall, all methods are better for small input standard deviations $\sigma$. The reason for this is that larger variances cause a larger part of the distribution to be mapped to 0 (for positive means) or to positive values (for negative means), which are the sources of error in the approximations. Results for additional architectures and non-linearities are presented in Supplementary Material C.1. There, we evaluate our method also for Leaky-ReLU, GELU [29], and SiLU [30].

## 4 LEARNING WITH UNCERTAINTY PROPAGATION

With the ability to propagate parametric distributions through neural networks, we can shift our focus to learning with such uncertainty propagation. Recall that the output of a neural network is now a (multivariate) Gaussian or a Cauchy distribution. We want to incorporate the covariances corresponding to a neural network's output into the loss function. This is simple to achieve for regression, i.e., by using log-likelihood as the loss, however, for classification it is more challenging.

### 4.1 REGRESSION

For regression (possibly with multiple outputs), we use the probability density of the predicted distribution at the corresponding value as our training objective. For normal distributions, our objective is maximizing the probability density by minimizing the negative log likelihood

$$\log \det(\boldsymbol{\Sigma}) + (\boldsymbol{y} - \boldsymbol{\mu})^\top \boldsymbol{\Sigma}^{-1}(\boldsymbol{y} - \boldsymbol{\mu}). \tag{6}$$

Here, the $k$ dimensional prediction is $\boldsymbol{\mu}$ with covariance matrix $\boldsymbol{\Sigma}$ and the ground truth value is $\boldsymbol{y}$. For Cauchy distributions, the respective probability density can be maximized analogously.

### 4.2 CLASSIFICATION

More interesting than regression is the classification case, for which we propose a new loss function. Previous work has proposed using moment matching of softmax [11], Dirichlet outputs [11], and an approximation to the $(n-1)$-variate logistic distribution [12]. Instead of finding a surrogate that incorporates the variance of the prediction, we use exact probabilities for classification. This allows incorporating not only the variances but also the covariances of a neural network's prediction.

To arrive at our loss, we compute the probability of correct classification, i.e., the probability that the score for a certain class is the maximum among all classes. As the exact probability of a score among $n \gg 2$ scores being the maximum is intractable, we resort to computing the pairwise exact probability of correct classification, which also allows us to consider their covariance.

The exact probability of pairwise correct classification for two random variables $X$ and $Y$ is

$$\mathbb{P}(X > Y) = \mathbb{P}(X - Y > 0) = \int_0^\infty \text{PDF}_{X-Y}(x)\,\mathrm{d}\,x = \text{CDF}_{Y-X}(0) \ . \tag{7}$$

For a multivariate normal distribution $(X, Y) \sim \mathcal{N}((\mu_X, \mu_Y), \boldsymbol{\Sigma})$ with covariance matrix $\Sigma$.

$$\mathbb{P}(X > Y) = \tfrac{1}{2}\left(1 + \text{erf}\left(\frac{\mu_X - \mu_Y}{\sqrt{2(\sigma_{XX}^2 + \sigma_{YY}^2 - 2\sigma_{XY})}}\right)\right) \tag{8}$$

For independent normal distributions $X \sim \mathcal{N}(\mu_X, \sigma_X^2)$ and $Y \sim \mathcal{N}(\mu_Y, \sigma_Y^2)$.

$$\mathbb{P}(X > Y) = \tfrac{1}{2}\left(1 + \text{erf}\left(\frac{\mu_X - \mu_Y}{\sqrt{2(\sigma_X^2 + \sigma_Y^2)}}\right)\right) \tag{9}$$

For Cauchy distributions $X \sim \mathcal{C}(x_X, \gamma_X)$ and $Y \sim \mathcal{C}(x_Y, \gamma_Y)$.

$$\mathbb{P}(X > Y) = \tfrac{1}{\pi}\left(\arctan\left(\frac{x_X - x_Y}{\gamma_X + \gamma_Y}\right)\right) + \tfrac{1}{2} \tag{10}$$

With that, our classification training objective is maximizing $\sum_{e \neq c} \frac{1}{k-1} \mathbb{P}(Z_c > Z_e)$, $\qquad$ (11) where $Z_c$ takes on the distribution of the correct class score and $Z_e$ takes on the distributions of the corresponding (among $k-1$) erroneous class scores. We refer to this loss function as the pairwise Gaussian and the pairwise Cauchy, respectively.

## 5 EXPERIMENTS

First, we investigate how well our method quantifies aleatoric uncertainty on UCI regression tasks. Second, we test how well our method can estimate epistemic uncertainty for detection of out-of-distribution test data. Third, we investigate robustness against Gaussian noise and adversarial attacks.

### 5.1 ALEATORIC UNCERTAINTY QUANTIFICATION

To evaluate aleatoric uncertainty predicted though uncertainty propagation, we examine its capability to produce calibrated prediction intervals on eight UCI regression tasks. We follow the experimental setting of Tagasovska *et al.* [6] and compare our method to their Conditional Quantiles method as well as to their best baseline, Conditional Gaussians [14]. Conditional Quantiles enable learning all conditional quantiles of a given target variable. Conditional Gaussian fits a conditional normal distribution, i.e., a neural network with two outputs, the mean and the variance. In comparison, our approach does not let a neural network predict the variance, but instead propagates input uncertainties through a neural network and computes calibrated prediction intervals from the output covariances.

We use the same model architecture, hyper-parameters, and evaluation metrics as Tagasovska *et al.* [6] for all three methods. The evaluation metrics are Prediction Interval Coverage Probability (PICP), i.e., the fraction of test data points falling into the predicted intervals, and the Mean Prediction Interval Width (MPIW). In Tab. 2, following Tagasovska *et al.* [6], we report the test PICP and MPIW of those models where the validation PICP lies between $92.5\%$ and $97.5\%$. The goal is to achieve a narrow interval (small MPIW) while the optimal test PICP is $95\%$. Our method achieves the narrowest well-calibrated prediction intervals. Specifically, for 5 out of 8 data sets, our method has the narrowest intervals while Conditional Gaussian as well as Conditional Quantile each achieve the narrowest well-calibrated prediction intervals on only 2 of the data sets. Notably, for the 'naval' task, our method has the best-calibrated coverage intervals while having by far the smallest MPIW.

Table 2: Results for the aleatoric uncertainty experiment. The task is to compute calibrated prediction intervals for 8 UCI data sets. Reported are test PICP and MPIW in parentheses. For MPIW lower is better. All results are averaged over 20 runs. Prior methods are duplicated from Tagasovska *et al.* [6].

| Data Set | Uncertainty Propagation | Conditional Gaussian | Conditional Quantile |
|---|---|---|---|
| concrete | $0.92 \pm 0.03$ (**0.25** $\pm 0.02$) | $0.94 \pm 0.03$ ($0.32 \pm 0.09$) | $0.94 \pm 0.03$ ($0.31 \pm 0.06$) |
| power | $0.94 \pm 0.01$ ($0.20 \pm 0.00$) | $0.94 \pm 0.01$ (**0.18** $\pm 0.00$) | $0.93 \pm 0.01$ (**0.18** $\pm 0.01$) |
| wine | $0.92 \pm 0.03$ (**0.45** $\pm 0.03$) | $0.94 \pm 0.02$ ($0.49 \pm 0.03$) | $0.93 \pm 0.03$ (**0.45** $\pm 0.04$) |
| yacht | $0.93 \pm 0.04$ ($0.06 \pm 0.01$) | $0.93 \pm 0.06$ (**0.03** $\pm 0.01$) | $0.93 \pm 0.06$ ($0.06 \pm 0.04$) |
| naval | $0.94 \pm 0.02$ (**0.02** $\pm 0.00$) | $0.96 \pm 0.01$ ($0.15 \pm 0.25$) | $0.95 \pm 0.02$ ($0.12 \pm 0.09$) |
| energy | $0.91 \pm 0.05$ (**0.05** $\pm 0.01$) | $0.94 \pm 0.03$ ($0.12 \pm 0.18$) | $0.94 \pm 0.03$ ($0.08 \pm 0.03$) |
| boston | $0.93 \pm 0.04$ (**0.28** $\pm 0.02$) | $0.94 \pm 0.03$ ($0.55 \pm 0.20$) | $0.92 \pm 0.06$ ($0.36 \pm 0.09$) |
| kin8nm | $0.95 \pm 0.01$ ($0.24 \pm 0.03$) | $0.93 \pm 0.01$ (**0.20** $\pm 0.01$) | $0.93 \pm 0.01$ ($0.23 \pm 0.02$) |

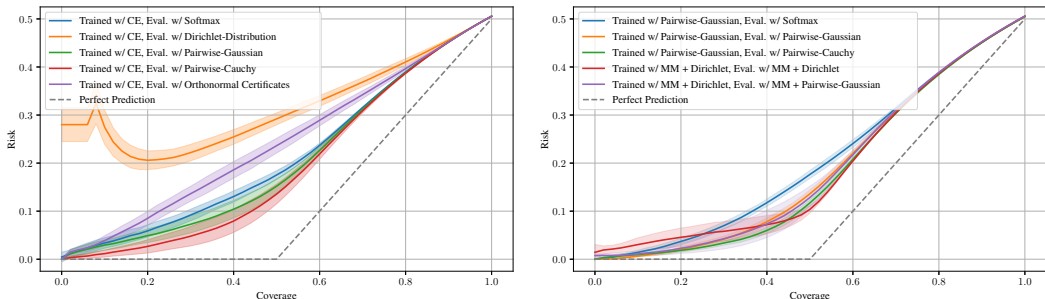

Figure 2: Selective prediction on MNIST with EMNIST letters as OOD data. Left: risk-coverage plots for off-the-shelf models trained with softmax cross-entropy. Right: models trained with uncertainty propagation. The grey line indicates perfect prediction. Results averaged over 10 runs.

Table 3: Selective prediction settings including the risk-coverage AUC for Fig. 2.

| Training Objective | Uncertainty Prop. | Selective Prediction | Risk-Coverage AUC |
| --- | --- | --- | --- |
| Softmax CE | — | Softmax Entropy | 21.4% |
| Softmax CE | Our Propagation | Pairwise Gaussian | 20.4% |
| Softmax CE | Our Propagation | Pairwise Cauchy | **19.2%** |
| Softmax CE | Our Propagation | Dirichlet Dist. | 32.0% |
| Softmax CE | — | Orthonormal Cert. | 24.2% |
| Pairwise Gaussian | Our Propagation | Softmax Entropy | 20.6% |
| Pairwise Gaussian | Our Propagation | Pairwise Gaussian | 19.0% |
| Pairwise Gaussian | Our Propagation | Pairwise Cauchy | **18.3%** |
| Dirichlet Dist. | Moment Matching | Dirichlet Dist. | 19.2% |
| Dirichlet Dist. | Moment Matching | Pairwise Gaussian | **19.0%** |
|  |  | Perfect Prediction | 12.5% |

## 5.2 EPISTEMIC UNCERTAINTY FOR OUT-OF-DISTRIBUTION SELECTIVE PREDICTION

Selective prediction [31] is a formulation where instead of only predicting a class, a neural network can also abstain from a prediction if it is not certain. We benchmark selective prediction on MNIST [32] and use EMNIST letters [33] as out-of-distribution data. EMNIST letters is a data set that contains letters from A to Z in the same format as MNIST. We train a neural network on the MNIST training data set and then combine the MNIST test data set (10 000 images) with 10 000 images from the EMNIST letter data set. This gives us a test data set of 20 000 images, 50% of which are out-of-distribution samples and for which the model should abstain from prediction.

In the first setting, we train a neural network with conventional softmax cross-entropy to simulate an existing off-the-shelf network. In the second setting, we train the neural network using uncertainty propagation with the pairwise Gaussian loss (ours) as well as using moment matching propagation and Dirichlet outputs [11]. We evaluate using five methods to compute certainty scores. First, we use the softmax entropy of the prediction, and apply Orthonormal Certificates from Tagasovska *et al.* [6]. Second, we propagate a distribution with $\sigma = 0.1$ through the network to obtain the covariances. Using these covariances, we use our pairwise Gaussian probabilities as well as the categorical probabilities of the Dirichlet outputs proposed by Gast *et al.* [11] to compute the entropies of the predictions. In Fig. 2, we provide risk-coverage plots [31] of the selective prediction based on these scores. Risk-coverage plots report the empirical risk (i.e., the error) for each degree of coverage $\alpha$. That is, we select the $\alpha$ most certain predictions and report the error. This corresponds to a setting, where a predictor can abstain from prediction in $1 - \alpha$ of the cases. We use risk-coverage area-under-the-curve (AUC) to quantify the overall selective prediction performance. Smaller AUC implies that the network is accurate on in-distribution test data, while abstaining from making a wrong prediction on out-of-distribution examples. In this experiment, no correct prediction on OOD data is possible because the classifier can only predict numbers while the OOD data consists of letters.

We evaluate a variety of combinations of training losses, uncertainty propagation methods, and confidence scores for selective prediction. See Tab. 3 for the summary and the corresponding risk-coverage AUC results. We find that training with an uncertainty-aware training objective (pairwise Gaussian

and Cauchy losses and Dirichlet outputs) improves the out-of-distribution detection substantially. Further, we find that the pairwise Cauchy scores achieve the best out-of-distribution detection on a pre-trained model. Orthonormal certificates (OC) estimate null-space of data in the latent space to detect out-of-distribution examples. In our setting, digits and letters may share a similar null-space, making OC not as effective. The Dirichtlet distribution does not perform well on a pretrained model because it is not designed for this scenario but instead designed to work in accordance with moment matching. We also observe that, for a model trained with Dirichlet outputs and moment matching uncertainty propagation, selective prediction confidence scores of pairwise Gaussian and pairwise Cauchy offer an improvement in comparison to the Dirichlet output based scores. This suggest that pairwise Cauchy scores are beneficial across various training approaches. The overall best accuracy (18.3% AUC) is achieved by training with an uncertainty-aware objective and evaluating using pairwise Cauchy. We also report AUC of a perfect predictor, i.e., one that always abstains on out-of-distribution data and always predicts correctly in-distribution.

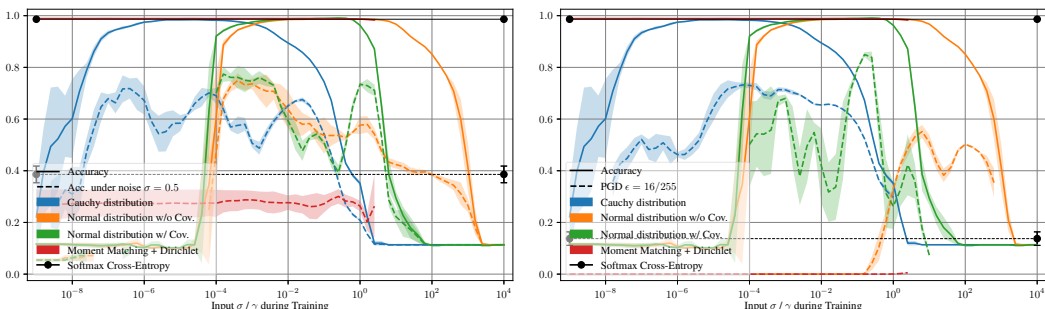

Figure 3: Robustness of CNNs on the MNIST data set against Gaussian noise (left) as well as PGD adversarial noise (right). The continuous lines show the test accuracy and the dashed lines show the robust accuracy. The black lines indicate the softmax cross-entropy trained baseline. Results are averaged over 3 runs. Results on CIFAR-10 and further settings are in Supplementary Material C.2.

## 5.3 ROBUSTNESS AGAINST GAUSSIAN AND ADVERSARIAL NOISE

Robustness of neural networks has received significant attention in the past years [34–36]. There are many methods for robustifying neural networks against random noise in the inputs or adversarially crafted perturbations. Our method is not explicitly designed for these purposes, however, here we empirically demonstrate that it achieves meaningful robustness improvements. This is an additional benefit to its main purpose of quantifying uncertainties.

We evaluate the robustness of classification models trained with uncertainty propagation on the MNIST [32] and CIFAR-10 [37] data sets. We consider the following training scenarios: First, we propagate normal distributions with and without covariance and train using the pairwise Gaussian loss. Second, we train with Cauchy distribution propagation using the pairwise Cauchy loss. As for baselines, we compare to training with softmax cross-entropy as well as moment matching propagation with the Dirichlet output loss [11]. The network architectures, hyper-parameters, as well as the results on CIFAR-10 are presented in Supplementary Material C.2.

**Random Gaussian Noise** To evaluate the random noise robustness of models trained with uncertainty propagation, we add random Gaussian per-pixel noise during evaluation. Specifically, we use input noise with $\sigma = 0.5$ (see Supplementary Material C.2 for $\sigma \in \{0.25, 0.75\}$) and clamp the image to the original pixel value range (between 0 and 1). In Fig. 3 (left), we display results for input standard deviations / scales from $10^{-9}$ to $10^4$. For every input standard deviation, we train separate models. For each method, the range of good hyperparameters is around $4 - 6$ orders of magnitude. We observe, when training with uncertainty propagation and the pairwise Gaussian / Cauchy losses, the models are substantially more robust than with conventional training on the MNIST data set.

**Adversarial Noise** To evaluate the adversarial robustness of models trained with uncertainty propagation, we use the projected gradient descent (PGD) attack [35] as it is a popular and powerful attack. In Fig. 3 (right), we demonstrate the adversarial robustness of CNNs on distributions. We can see that our approach is competitive with the accuracy of the softmax cross-entropy loss and can even outperform it. For measuring the adversarial robustness, we use $L_\infty$-bounded PGD, for

MNIST with $\epsilon = 16/255$ (and with $\epsilon = 8/255$ in the supplementary material), and for CIFAR with $\epsilon \in \{3/255, 4/255\}$ (results for CIFAR are presented in the supplementary material). Note that our uncertainty propagation, in combination with pairwise probability classification losses, outperform both baselines in each case. We find that the adversarially most robust training method is propagating normal distributions with covariances, where we achieve a robust accuracy of $85\%$ with $\epsilon = 16/255$ on the MNIST data set compared to a robust accuracy of $14\%$ for models trained with softmax cross-entropy. Note that the baseline Dirichlet loss with moment-matching [11] offers no robustness gains in this experiment. We observe that, when training with normal distributions without covariances, the models are only robust against adversarial noise for rather large input standard deviations. At the same time, this effect is inverse for random Gaussian noise, where robustness against random noise is achieved when input standard deviations are rather small.

We perform PGD on our models with our objective functions as described in Eq. 11 and PGD on the cross-entropy trained models with the softmax cross-entropy objective. In Supplementary Material C.2, we provide results where a softmax cross-entropy based objective is used by the adversary: our models are more robust against the cross-entropy based attack than against the true objective based attacks which are reported in the plot. Our methods are not as robust as methods specifically designed for that purpose, e.g., adversarial training [35]. However, our methods provide a substantial improvement over training with softmax cross-entropy.

### 5.4 Propagating Distributions through ResNets

To validate that our method also performs well for large architectures, we apply it to learning CIFAR-10 with ResNet-18 and ResNet-34 [38] architectures. Here, we train each model using Adam [39] for 400 epochs and compare it to the same setting with the softmax cross-entropy loss

Table 4: CIFAR-10 performance with ResNets.

| Model | Softmax CE | Pairwise Gaussian (w/ cov.) |
|---|---|---|
| ResNet-18 | $90.1\% \pm 0.2\%$ | $89.5\% \pm 0.6\%$ |
| ResNet-34 | $90.5\% \pm 0.2\%$ | $89.6\% \pm 0.2\%$ |

in Tab. 4. The results validate that training with the Pairwise Gaussian loss (w/ covariances) does not substantially damage the performance in comparsion to softmax cross-entropy.

### 5.5 Runtime Analysis

In Tab. 5, we report runtimes for propagating distributions. For small models, propagating distributions without covariances is almost without overhead while propagating covariances is slower. Moment matching is slightly more expensive than the proposed propagation, which is because moment matching requires evaluating additional (scalar) functions at the non-linearities. The complexity (as a factor of the cost of propagating

Table 5: Computational Cost Benchmark. Times per epoch on CIFAR-10 with a batch size of 128 on a single V100 GPU.

| Model | regular | moment matching | dist. w/o cov. | dist. w/ cov. |
|---|---|---|---|---|
| 3 Layer CNN | 6.29s | 6.33s | 6.31s | 20.8s |
| ResNet-18 | 9.85s | 30.7s* | 28.5s | 251s |

\* as moment matching for some layers of ResNet (such as MaxPool for more than two inputs) is very expensive and does not have a closed form solution in the literature, we use our approximation for these layers.

a single point through the neural network) of propagating distributions with covariances is linear in the number of outputs, while for propagating distributions without covariances it is constant. As a reference, exact propagation of the distributions (and not a parametric approximation as in this work) has an exponential runtime factor in the number of neurons [40]. The runtime factor of DVIA [27] is linear in the number of neurons in the largest layer, which makes experiments exceeding Table 1 infeasible. For example, in the robustness experiments on MNIST dataset in Section 5.3, the dimension of the largest hidden layer is $\approx 12\,500$, i.e., computational time of a single forward pass with DVIA is over $12\,500$ regular forward passes.

## 6 Conclusion

We studied the problem of uncertainty propagation in neural networks. Uncertainty propagation allows quantifying both epistemic and aleatoric uncertainties, and promotes robustness to input noises, as demonstrated in our empirical studies. Our local linearization based methods, together with the pairwise Gaussian and Cauchy loss functions, performed well. Extending our theoretical results to better understand the empirical success could be an interesting direction for future work.

REPRODUCIBILITY STATEMENT

We will make the source code and experiments of this work publicly available to foster future research in this direction. All data sets are publicly available. We specify all necessary hyperparameters for each experiment. Each experiment can be reproduced on a single GPU.

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

# Supplementary Material for
# Propagating Distributions through Neural Networks

## A ILLUSTRATION OF PARAMETRIC APPROXIMATIONS OF ReLU

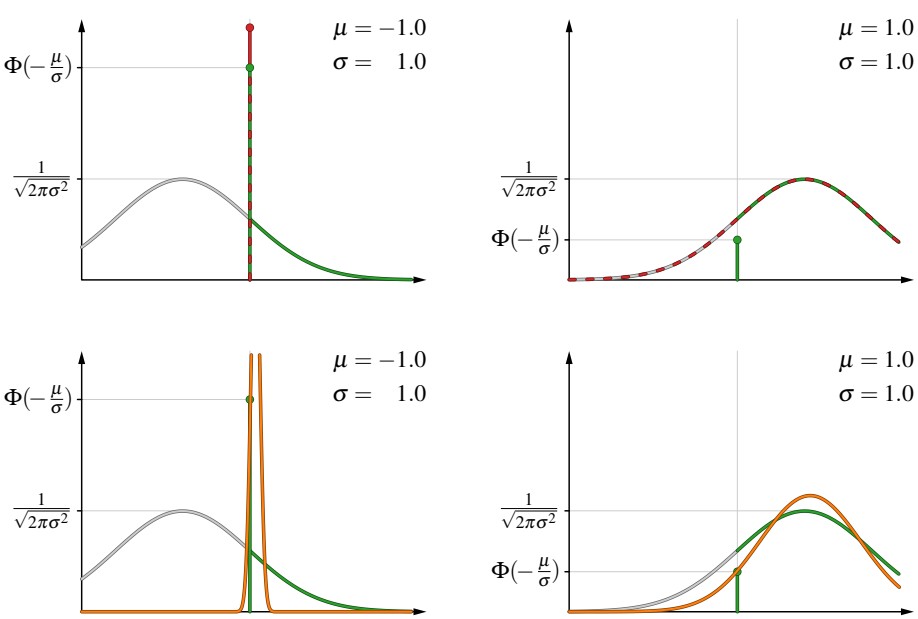

Figure 4: Illustration of parametric approximations of ReLU. In each plot, the gray distribution is the input and the green distribution is the true output distribution. The red distributions (top) are our local linearization approximation. The orange distributions (bottom) are produced by moment matching.

## B TRANSFORMATION OF CONVOLUTIONS

For two-dimensional convolutions, we use the notation $\boldsymbol{Y} = \boldsymbol{X} * \mathbf{W} + \boldsymbol{B}$ where $\boldsymbol{X} \in \mathbb{R}^{c_0 \times n_0 \times n_1}$ is the input (image), $\mathbf{W} \in \mathbb{R}^{c_1 \times c_0 \times k_0 \times k_1}$ is the weight tensor, and $\boldsymbol{B}, \boldsymbol{Y} \in \mathbb{R}^{c_1 \times m_0 \times m_1}$ are the bias and output tensors with $c_0, c_1, k_0, k_1, n_0, n_1, m_0, m_1, m_0, m_1 \in \mathbb{N}_+$. Further, let $\boldsymbol{X}^{\mathsf{U}} \in \mathbb{R}^{m_0 \times m_1 \times c_0 \times k_0 \times k_1}$ be the unfolded sliding local blocks[1] from tensor $\boldsymbol{X}$ such that $\boldsymbol{Y}_{c_1 m_0 m_1} = \sum_{c_0 k_0 k_1} \boldsymbol{X}^{\mathsf{U}}_{m_0 m_1 c_0 k_0 k_1} \mathbf{W}_{c_1 c_0 k_0 k_1} + \boldsymbol{B} = \boldsymbol{X} * \mathbf{W} + \boldsymbol{B}$.

**Normal Distribution with Covariances** For the convolutional layer, $\boldsymbol{\mu} \in \mathbb{R}^{c_0 \times n_0 \times n_1}, \boldsymbol{\Sigma} \in \mathbb{R}^{c_0 \times n_0 \times n_1 \times c_0 \times n_0 \times n_1}$. Note that $\boldsymbol{\Sigma}^{\mathsf{U}} \in \mathbb{R}^{m_0 \times m_1 \times c_0 \times k_0 \times k_1 \times m_0 \times m_1 \times c_0 \times k_0 \times k_1}$ is the unfolded sliding local blocks covariance of the covariance tensor $\boldsymbol{\Sigma}$. $\boldsymbol{\mu} \mapsto \boldsymbol{\mu} * \mathbf{W} + \boldsymbol{B}$ and $\boldsymbol{\Sigma}_{c_1 m_0 m_1 c_1' m_0' m_1'} \mapsto \sum_{c_0 k_0 k_1 c_0' k_0' k_1'} \mathbf{W}_{c_1 c_0 k_0 k_1} \boldsymbol{\Sigma}^{\mathsf{U}}_{m_0 m_1 c_0 k_0 k_1 m_0' m_1' c_0' k_0' k_1'} \mathbf{W}_{c_1' c_0' k_0' k_1'}$.

**Normal Distribution without Covariances** For the convolutional layer, $\boldsymbol{\mu}, \boldsymbol{\sigma} \in \mathbb{R}^{c_0 \times n_0 \times n_1}$. $\boldsymbol{\mu} \mapsto \boldsymbol{\mu} * \mathbf{W} + \boldsymbol{B}$ and $\boldsymbol{\sigma}^2 \mapsto \boldsymbol{\sigma}^2 * \mathbf{W}^2$.

**Cauchy Distribution** For the convolutional layer, $\boldsymbol{x}_0, \boldsymbol{\gamma} \in \mathbb{R}^{c_0 \times n_0 \times n_1}$. $\boldsymbol{x}_0 \mapsto \boldsymbol{x}_0 * \mathbf{W} + \boldsymbol{B}$ and $\boldsymbol{\gamma} \mapsto \boldsymbol{\gamma} * \mathrm{Abs}(\mathbf{W})$.

---

[1]Einstein summation notation. The unfolded sliding local blocks are equivalent to `torch.unfold`.

# C  ADDITIONAL EXPERIMENTS

## C.1  SIMULATION OF PROPAGATING NORMAL DISTRIBUTIONS

Tab. 6–22 show additional simulations with 1, 2, 4, and 6 hidden layers as well as ReLU, Leaky-ReLU, GELU, SiLU, and logistic sigmoid activations. Here, the reported metric is intersection of probability mass, i.e., $1 - TV$. The baseline is computed via $10^6$ samples.

Tab. 23 considers the average ratio between predicted and true standard deviations.

As there we did not find a moment matching method for GELU and SiLU in the literature, we omit moment matching in these cases. DVIA [27] is only applicable to ReLU among the non-linearities we consider. For moment matching with logistic sigmoid, we use numerical integration as it does not have a closed form solution.

Table 6: Simulation of propagating normal distributions. The network is a **2** layer ReLU activated network with dimensions `4-100-3`, i.e., **1 ReLU** activation.

| $\sigma$ | With Covariances | DVIA [27] | Without Covariances | Moment Matching |
|---|---|---|---|---|
| 0.1 | $0.9874 \pm 0.0079$ | $0.9875 \pm 0.0074$ | $0.4574 \pm 0.0275$ | $0.4552 \pm 0.0288$ |
| 1 | $0.9578 \pm 0.0188$ | $0.9619 \pm 0.0165$ | $0.4643 \pm 0.0235$ | $0.4439 \pm 0.0238$ |
| 10 | $0.8648 \pm 0.0206$ | $0.8982 \pm 0.0247$ | $0.4966 \pm 0.0232$ | $0.4580 \pm 0.0136$ |
| 100 | $0.8157 \pm 0.0231$ | $0.8608 \pm 0.0353$ | $0.5034 \pm 0.0248$ | $0.4640 \pm 0.0178$ |
| 1000 | $0.8103 \pm 0.0236$ | $0.8555 \pm 0.0372$ | $0.5041 \pm 0.0254$ | $0.4640 \pm 0.0182$ |

Table 7: Simulation of propagating normal distributions. The network is a **3** layer ReLU activated network with dimensions `4-100-100-3`, i.e., **2 ReLU** activations.

| $\sigma$ | With Covariances | DVIA [27] | Without Covariances | Moment Matching |
|---|---|---|---|---|
| 0.1 | $0.9861 \pm 0.0099$ | $0.9840 \pm 0.0112$ | $0.3070 \pm 0.0185$ | $0.3111 \pm 0.0193$ |
| 1 | $0.9259 \pm 0.0279$ | $0.9247 \pm 0.0270$ | $0.3123 \pm 0.0133$ | $0.3004 \pm 0.0136$ |
| 10 | $0.8093 \pm 0.0234$ | $0.8276 \pm 0.0255$ | $0.3463 \pm 0.0206$ | $0.2972 \pm 0.0170$ |
| 100 | $0.7439 \pm 0.0257$ | $0.8152 \pm 0.0347$ | $0.3931 \pm 0.0207$ | $0.3065 \pm 0.0237$ |
| 1000 | $0.7373 \pm 0.0259$ | $0.8061 \pm 0.0384$ | $0.3981 \pm 0.0188$ | $0.3079 \pm 0.0249$ |

Table 8: Simulation of propagating normal distributions. The network is a **5** layer ReLU activated network with dimensions `4-100-100-100-100-3`, i.e., **4 ReLU** activations.

| $\sigma$ | With Covariances | DVIA [27] | Without Covariances | Moment Matching |
|---|---|---|---|---|
| 0.1 | $0.9791 \pm 0.0202$ | $0.9720 \pm 0.0228$ | $0.2361 \pm 0.0250$ | $0.2611 \pm 0.0256$ |
| 1 | $0.8747 \pm 0.0546$ | $0.8519 \pm 0.0543$ | $0.2243 \pm 0.0237$ | $0.2195 \pm 0.0294$ |
| 10 | $0.7586 \pm 0.0407$ | $0.7035 \pm 0.0366$ | $0.2186 \pm 0.0244$ | $0.1976 \pm 0.0179$ |
| 100 | $0.6877 \pm 0.0333$ | $0.5845 \pm 0.0479$ | $0.2261 \pm 0.0282$ | $0.1724 \pm 0.0111$ |
| 1000 | $0.6808 \pm 0.0318$ | $0.5318 \pm 0.0516$ | $0.2193 \pm 0.0248$ | $0.1706 \pm 0.0109$ |

Table 9: Simulation of propagating normal distributions. The network is a **7** layer ReLU activated network with dimensions `4-100-100-100-100-100-100-3`, i.e., **6 ReLU** activations.

| $\sigma$ | With Covariances | DVIA [27] | Without Covariances | Moment Matching |
|---|---|---|---|---|
| 0.1 | $0.9732 \pm 0.0219$ | $0.9660 \pm 0.0226$ | $0.2196 \pm 0.0208$ | $0.2601 \pm 0.0323$ |
| 1 | $0.8494 \pm 0.0853$ | $0.8166 \pm 0.0986$ | $0.2292 \pm 0.0303$ | $0.2236 \pm 0.0368$ |
| 10 | $0.7743 \pm 0.0477$ | $0.6309 \pm 0.0553$ | $0.2420 \pm 0.0295$ | $0.2327 \pm 0.0571$ |
| 100 | $0.7077 \pm 0.0348$ | $0.5265 \pm 0.0975$ | $0.3525 \pm 0.2337$ | $0.1800 \pm 0.0188$ |
| 1000 | $0.7013 \pm 0.0334$ | $0.5166 \pm 0.1223$ | $0.4422 \pm 0.2450$ | $0.1764 \pm 0.0175$ |

Table 10: Simulation of propagating normal distributions. The network is a **2** layer Leaky-ReLU activated network with dimensions `4-100-3`, i.e., **1 Leaky-ReLU** activation with negative slope $\alpha = 0.1$.

| $\sigma$ | With Covariances | Without Covariances | Moment Matching |
|---|---|---|---|
| 0.1 | $0.9852 \pm 0.0065$ | $0.4484 \pm 0.0204$ | $0.4481 \pm 0.0208$ |
| 1 | $0.9624 \pm 0.0152$ | $0.4547 \pm 0.0183$ | $0.4417 \pm 0.0172$ |
| 10 | $0.8876 \pm 0.0194$ | $0.4838 \pm 0.0198$ | $0.4454 \pm 0.0118$ |
| 100 | $0.8458 \pm 0.0226$ | $0.4912 \pm 0.0203$ | $0.4581 \pm 0.0139$ |
| 1000 | $0.8411 \pm 0.0232$ | $0.4918 \pm 0.0207$ | $0.4594 \pm 0.0141$ |

Table 11: Simulation of propagating normal distributions. The network is a **3** layer Leaky-ReLU activated network with dimensions `4-100-100-3`, i.e., **2 Leaky-ReLU** activations with negative slope $\alpha = 0.1$.

| $\sigma$ | With Covariances | Without Covariances | Moment Matching |
|---|---|---|---|
| 0.1 | $0.9806 \pm 0.0085$ | $0.3020 \pm 0.0162$ | $0.3063 \pm 0.0171$ |
| 1 | $0.9324 \pm 0.0255$ | $0.3055 \pm 0.0129$ | $0.3007 \pm 0.0124$ |
| 10 | $0.8321 \pm 0.0254$ | $0.3343 \pm 0.0180$ | $0.2963 \pm 0.0147$ |
| 100 | $0.7716 \pm 0.0294$ | $0.3769 \pm 0.0203$ | $0.3040 \pm 0.0189$ |
| 1000 | $0.7651 \pm 0.0297$ | $0.3825 \pm 0.0190$ | $0.3056 \pm 0.0198$ |

Table 12: Simulation of propagating normal distributions. The network is a **5** layer Leaky-ReLU activated network with dimensions `4-100-100-100-100-3`, i.e., **4 Leaky-ReLU** activations with negative slope $\alpha = 0.1$.

| $\sigma$ | With Covariances | Without Covariances | Moment Matching |
|---|---|---|---|
| 0.1 | $0.9689 \pm 0.0163$ | $0.2325 \pm 0.0224$ | $0.2566 \pm 0.0249$ |
| 1 | $0.8856 \pm 0.0460$ | $0.2260 \pm 0.0197$ | $0.2218 \pm 0.0236$ |
| 10 | $0.7638 \pm 0.0393$ | $0.2146 \pm 0.0220$ | $0.1975 \pm 0.0169$ |
| 100 | $0.6912 \pm 0.0352$ | $0.2364 \pm 0.0227$ | $0.1756 \pm 0.0112$ |
| 1000 | $0.6835 \pm 0.0337$ | $0.2361 \pm 0.0216$ | $0.1743 \pm 0.0116$ |

Table 13: Simulation of propagating normal distributions. The network is a **7** layer Leaky-ReLU activated network with dimensions `4-100-100-100-100-100-100-3`, i.e., **6 Leaky-ReLU** activations with negative slope $\alpha = 0.1$.

| $\sigma$ | With Covariances | Without Covariances | Moment Matching |
|---|---|---|---|
| 0.1 | $0.9585 \pm 0.0203$ | $0.2197 \pm 0.0199$ | $0.2519 \pm 0.0292$ |
| 1 | $0.8573 \pm 0.0720$ | $0.2251 \pm 0.0289$ | $0.2220 \pm 0.0351$ |
| 10 | $0.7701 \pm 0.0465$ | $0.2332 \pm 0.0301$ | $0.2243 \pm 0.0259$ |
| 100 | $0.7047 \pm 0.0378$ | $0.2940 \pm 0.1864$ | $0.1779 \pm 0.0171$ |
| 1000 | $0.6979 \pm 0.0371$ | $0.3518 \pm 0.2209$ | $0.1746 \pm 0.0159$ |

Table 14: Simulation of propagating normal distributions. The network is a **2** layer SiLU activated network with dimensions `4-100-3`, i.e., **1 SiLU** activation.

| $\sigma$ | With Covariances | Without Covariances |
|---|---|---|
| 0.1 | $0.9910 \pm 0.0025$ | $0.4494 \pm 0.0192$ |
| 1 | $0.9813 \pm 0.0076$ | $0.4550 \pm 0.0158$ |
| 10 | $0.8635 \pm 0.0180$ | $0.5158 \pm 0.0251$ |
| 100 | $0.8104 \pm 0.0201$ | $0.5328 \pm 0.0282$ |
| 1000 | $0.8050 \pm 0.0205$ | $0.5347 \pm 0.0288$ |

Table 18: Simulation of propagating normal distributions. The network is a **2** layer GELU activated network with dimensions `4-100-3`, i.e., **1 GELU** activation.

| $\sigma$ | With Covariances | Without Covariances |
|---|---|---|
| 0.1 | $0.9875 \pm 0.0075$ | $0.4567 \pm 0.0317$ |
| 1 | $0.9732 \pm 0.0096$ | $0.4682 \pm 0.0225$ |
| 10 | $0.8530 \pm 0.0185$ | $0.5204 \pm 0.0266$ |
| 100 | $0.8039 \pm 0.0211$ | $0.5251 \pm 0.0278$ |
| 1000 | $0.7987 \pm 0.0216$ | $0.5255 \pm 0.0282$ |

Table 15: Simulation of propagating normal distributions. The network is a **3** layer SiLU activated network with dimensions `4-100-100-3`, i.e., **2 SiLU** activations.

| $\sigma$ | With Covariances | Without Covariances |
|---|---|---|
| 0.1 | $0.9876 \pm 0.0051$ | $0.3056 \pm 0.0178$ |
| 1 | $0.9658 \pm 0.0121$ | $0.3076 \pm 0.0122$ |
| 10 | $0.8067 \pm 0.0136$ | $0.3531 \pm 0.0189$ |
| 100 | $0.7425 \pm 0.0162$ | $0.3806 \pm 0.0261$ |
| 1000 | $0.7365 \pm 0.0167$ | $0.3829 \pm 0.0233$ |

Table 19: Simulation of propagating normal distributions. The network is a **3** layer GELU activated network with dimensions `4-100-100-3`, i.e., **2 GELU** activations.

| $\sigma$ | With Covariances | Without Covariances |
|---|---|---|
| 0.1 | $0.9860 \pm 0.0054$ | $0.3065 \pm 0.0244$ |
| 1 | $0.9453 \pm 0.0169$ | $0.3129 \pm 0.0164$ |
| 10 | $0.8024 \pm 0.0128$ | $0.3560 \pm 0.0176$ |
| 100 | $0.7446 \pm 0.0155$ | $0.3966 \pm 0.0237$ |
| 1000 | $0.7390 \pm 0.0159$ | $0.4004 \pm 0.0222$ |

Table 16: Simulation of propagating normal distributions. The network is a **5** layer SiLU activated network with dimensions `4-100-100-100-100-3`, i.e., **4 SiLU** activations.

| $\sigma$ | With Covariances | Without Covariances |
|---|---|---|
| 0.1 | $0.9704 \pm 0.0126$ | $0.2471 \pm 0.0277$ |
| 1 | $0.9174 \pm 0.0343$ | $0.2414 \pm 0.0096$ |
| 10 | $0.7478 \pm 0.0342$ | $0.2357 \pm 0.0141$ |
| 100 | $0.6846 \pm 0.0241$ | $0.2453 \pm 0.0254$ |
| 1000 | $0.6789 \pm 0.0238$ | $0.2371 \pm 0.0236$ |

Table 20: Simulation of propagating normal distributions. The network is a **5** layer GELU activated network with dimensions `4-100-100-100-100-3`, i.e., **4 GELU** activations.

| $\sigma$ | With Covariances | Without Covariances |
|---|---|---|
| 0.1 | $0.9706 \pm 0.0104$ | $0.2389 \pm 0.0287$ |
| 1 | $0.8916 \pm 0.0359$ | $0.2442 \pm 0.0131$ |
| 10 | $0.7607 \pm 0.0247$ | $0.2460 \pm 0.0220$ |
| 100 | $0.6975 \pm 0.0184$ | $0.2532 \pm 0.0295$ |
| 1000 | $0.6915 \pm 0.0193$ | $0.2465 \pm 0.0288$ |

Table 17: Simulation of propagating normal distributions. The network is a **7** layer SiLU activated network with dimensions `4-100-100-100-100-100-100-3`, i.e., **6 SiLU** activations.

| $\sigma$ | With Covariances | Without Covariances |
|---|---|---|
| 0.1 | $0.9147 \pm 0.0421$ | $0.2326 \pm 0.0345$ |
| 1 | $0.8248 \pm 0.0777$ | $0.2348 \pm 0.1062$ |
| 10 | $0.7293 \pm 0.0388$ | $0.2762 \pm 0.1315$ |
| 100 | $0.6726 \pm 0.0411$ | $0.2326 \pm 0.0282$ |
| 1000 | $0.6675 \pm 0.0417$ | $0.2363 \pm 0.0309$ |

Table 21: Simulation of propagating normal distributions. The network is a **7** layer GELU activated network with dimensions `4-100-100-100-100-100-100-3`, i.e., **6 GELU** activations.

| $\sigma$ | With Covariances | Without Covariances |
|---|---|---|
| 0.1 | $0.9262 \pm 0.0294$ | $0.2261 \pm 0.0352$ |
| 1 | $0.8387 \pm 0.0858$ | $0.2710 \pm 0.1547$ |
| 10 | $0.7342 \pm 0.0266$ | $0.3557 \pm 0.2025$ |
| 100 | $0.6898 \pm 0.0218$ | $0.2244 \pm 0.0259$ |
| 1000 | $0.6854 \pm 0.0220$ | $0.2235 \pm 0.0264$ |

Table 22: Simulation of propagating normal distributions. The network is a **5** layer Logistic Sigmoid activated network with dimensions `4-100-100-100-100-3`, i.e., **4 Logistic Sigmoid** activations.

| $\sigma$ | With Covariances | Without Covariances | Moment Matching |
|---|---|---|---|
| 0.1 | 0.9890 | 0.7809 | 0.7793 |
| 1 | 0.9562 | 0.7880 | 0.7912 |
| 10 | 0.8647 | 0.8656 | 0.7674 |
| 100 | 0.8443 | 0.8442 | 0.8027 |
| 1000 | 0.8440 | 0.8440 | 0.8070 |

Table 23: Simulation of propagating normal distributions. The network is a **5** layer ReLU activated network with dimensions `4-100-100-100-100-3`, i.e., **4 ReLU** activations. Displayed is the average ratio between the output standard deviations. The 3 values, correspond to the three output dimensions of the Iris model.
A value of 1 is optimal. We find that both our method with covariances as well as DVIA achieve a good accuracy in this setting, while the methods which do not consider covariances (ours (w/o cov.) and Moment Matching) underestimate the output standard deviations by a large factor. For small input standard deviations, our method w/ cov. as well as DVIA perform better. For large input standard deviations, our method tends to overestimate the output standard deviation, while DVIA underestimates the standard deviation. The ratios for our method and DVIA (while going into opposite directions) are similar, e.g., 2.6095 for our method and 1/0.3375=2.9629 for DVIA.

| $\sigma$ | With Cov. | | | DVIA | | | Without Cov. | | | Moment Matching | | |
|---|---|---|---|---|---|---|---|---|---|---|---|---|
| 0.1 | 0.9933 | 1.0043 | 0.9954 | 0.9904 | 0.9917 | 0.9906 | 0.0126 | 0.0180 | 0.0112 | 0.0255 | 0.0373 | 0.0211 |
| 1 | 0.9248 | 1.0184 | 0.9598 | 0.8462 | 0.9256 | 0.8739 | 0.0103 | 0.0170 | 0.0100 | 0.0101 | 0.0166 | 0.0097 |
| 10 | 1.9816 | 2.2526 | 2.0629 | 0.6035 | 0.6664 | 0.6172 | 0.0207 | 0.0359 | 0.0205 | 0.0166 | 0.0269 | 0.0157 |
| 100 | 2.5404 | 2.8161 | 2.6217 | 0.3721 | 0.4194 | 0.3849 | 0.0264 | 0.0450 | 0.0260 | 0.0186 | 0.0296 | 0.0174 |
| 1000 | 2.6095 | 2.8857 | 2.6906 | 0.3375 | 0.3839 | 0.3503 | 0.0271 | 0.0462 | 0.0267 | 0.0188 | 0.0298 | 0.0175 |

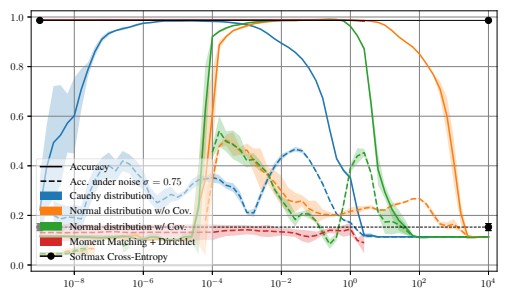 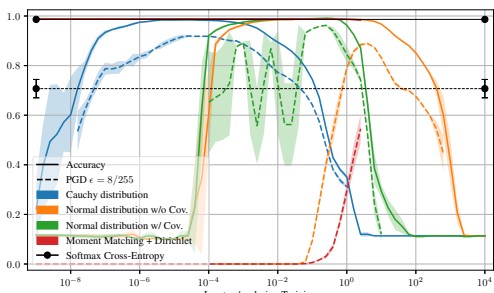

Figure 5: Robustness of CNNs on the MNIST data set against Gaussian noise (left) as well as PGD adversarial noise (right). While Fig. 3 presents $\sigma = 0.5$ and $\epsilon = 16/255$, this figure presents $\sigma = 0.75$ and $\epsilon = 8/255$ The continuous lines show the test accuracy and the dashed lines show the robust accuracy. The black lines indicate the softmax cross-entropy trained baseline. Results are averaged over 3 runs.

## C.2 GAUSSIAN AND ADVERSARIAL NOISE ROBUSTNESS

In Fig. 5, we present an extension to Fig. 3 which demonstrates random and adversarial noise robustness for alternative noise intensities.

In Fig. 9 and 10, we present an additional demonstration of the adversarial and random noise robustness of CNNs on distributions. Note that we do not include results for normal distributions with covariances (as in Figures 3 and 5). Here, the emphasis is set on using both MNIST and CIFAR.

In Fig. 6, we show the robustness where a cross-entropy based objective (green) is used by the adversary. A cross-entropy based objective could be used in practice in multiple scenarios: For example, publishing just the weights of the model without information about the objective function, or by using a standard library for adversarial attacks that does not require or support specification of the training objective. We emphasize that our models are more robust against the cross-entropy based attack (CE-PGD) than the pairwise distribution loss based attack (PA-PGD) which is reported in all other plots.

Note that the phenomenon of vanishing gradients can occur in our models (as it can also occur in any cross-entropy based model). Thus, we decided to consider all attempted attacks, where the gradient is zero or zero for some of the pixels in the input image, as successful attacks because the attack might be mitigated by a lack of gradients. In Fig. 7, we compare conventional robustness to robustness with the additional Gradient-Non-Zero (GNZ) requirement. We emphasize that *we use the GNZ requirement also in all other plots.*

For evaluating the robustness against random noise, we use input noise with $\sigma \in \{0.25, 0.5, 0.75\}$ and clamp the image to the original pixel value range.

## C.3 PROPAGATING WITHOUT COVARIANCES

We analyze the accuracy of a fully connected neural network (FCNN) where a normal distribution is introduced in the $k$th out of 5 layers. Here, the normal distribution is modeled without covariances. The result is displayed in Fig. 8, where we can observe that there is an optimal scale such that the accuracy is as large as possible. Further, we can see the behavior if the distribution is not introduced in the first layer, but in a later layer instead. That is, the first layers are conventional layers and the distribution is introduced after a certain number of layers. Here, we can see that the more layers propagate distributions rather than single values, the larger the optimal standard deviation is. The reason for this is that the scale of the distribution decays with each layer such that for deep models a larger input standard deviation is necessary to get an adequate output standard deviation for training the model. This is because we modeled the distributions without covariances, which demonstrates that propagating without covariances underestimates the variances.

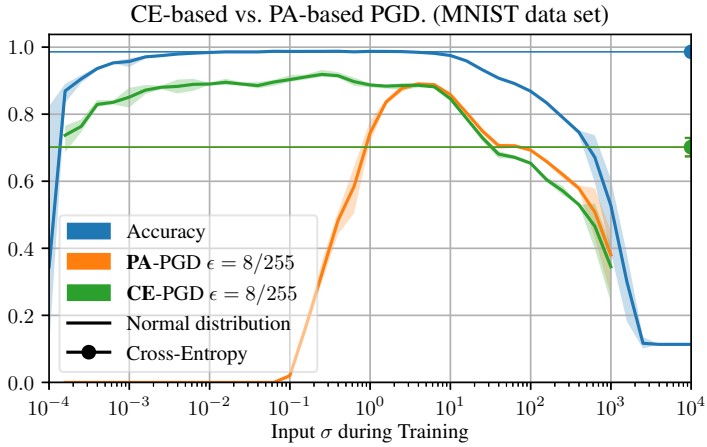

Figure 6: Softmax Cross-Entropy (CE) vs. pairwise distribution loss (PA) based PGD attack. Averaged over 3 runs.

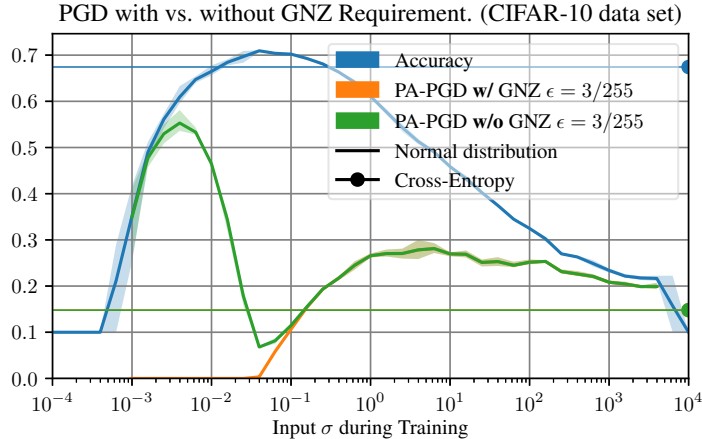

Figure 7: Effect of the gradient-non-zero (GNZ) requirement. Averaged over 3 runs.

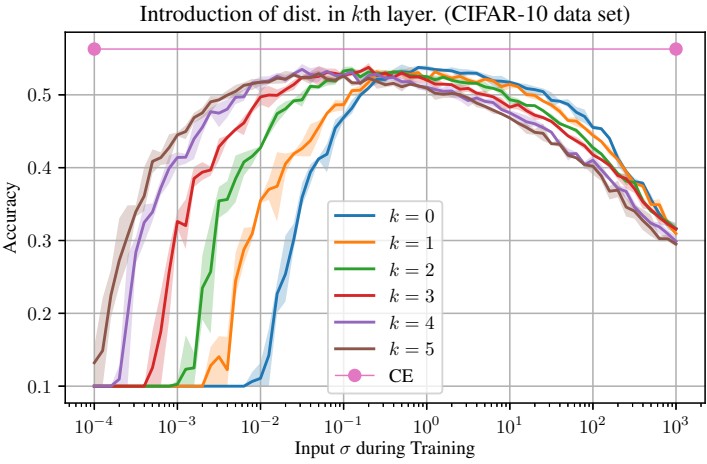

Figure 8: The normal distribution is introduced in the $k$th layer of a 5 layer fully connected network ($k \in \{0, 1, 2, 3, 4, 5\}$). Averaged over 3 runs.

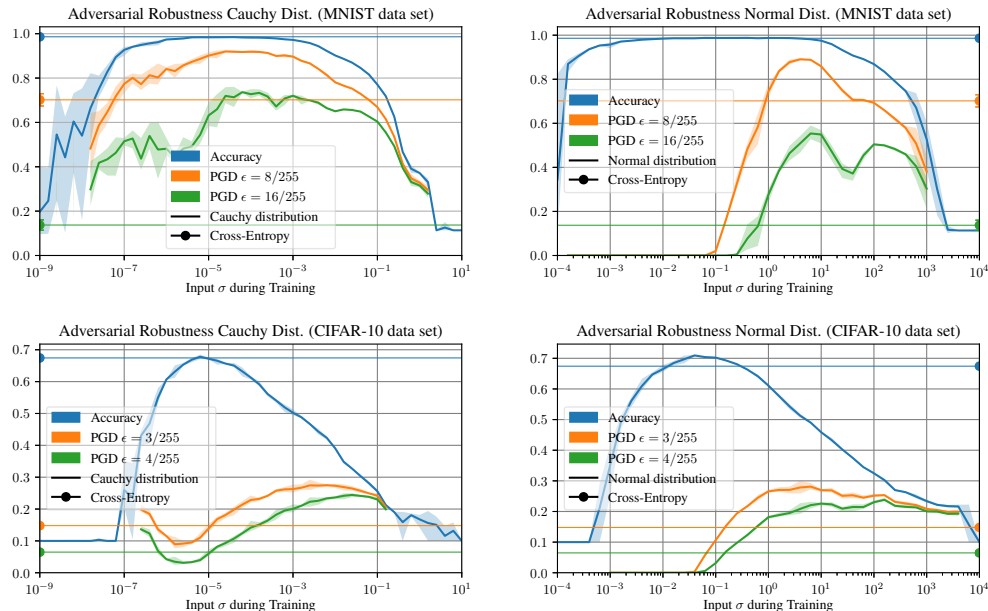

Figure 9: Accuracy and adversarial robustness under PGD attack. Left: Cauchy distribution. Right: Normal distribution *without* propagating covariances. Top: MNIST data set. Bottom: CIFAR-10 data set. Averaged over 3 runs.

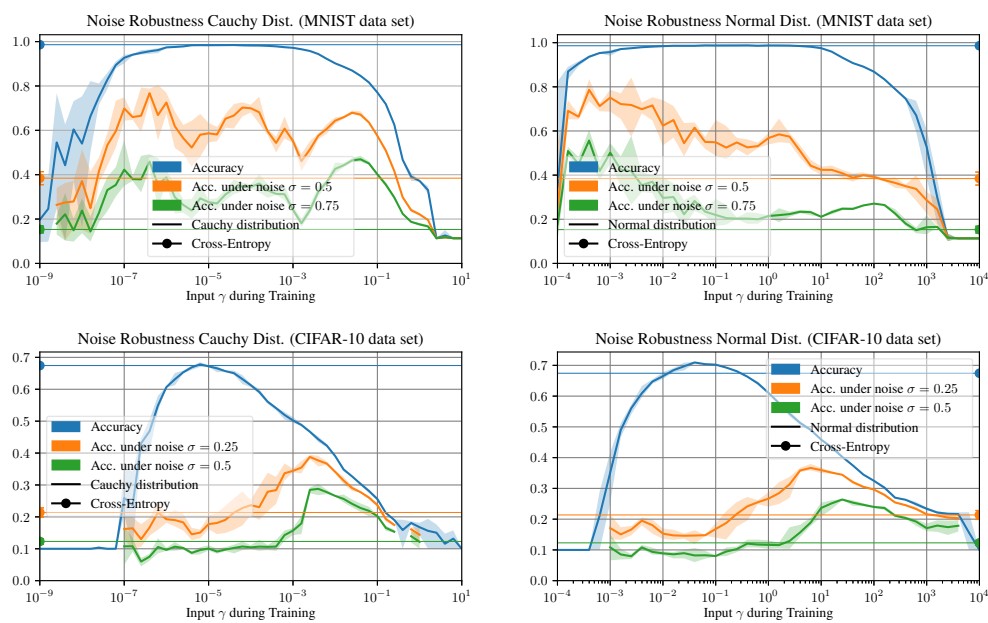

Figure 10: Accuracy and robustness against random normal noise. Left: Cauchy distribution. Right: Normal distribution *without* propagating covariances. Top: MNIST data set. Bottom: CIFAR-10 data set. Averaged over 3 runs.

## D    IMPLEMENTATION DETAILS

### D.1    SIMULATION OF PROPAGATING NORMAL DISTRIBUTIONS

We trained a fully-connected neural network with 1, 2, or 4 hidden layers and ReLU / Leaky-ReLU activations for 5000 epochs on the Iris data set via the softmax cross-entropy loss. Here, each hidden layer has 100 hidden neurons. Results are averaged over 10 runs, and 10 inputs are propagated for each of these 10 runs. Across all methods of propagating distributions, each Monte Carlo baseline as well as model weights are shared.

### D.2    ALEATORIC UNCERTAINTY QUANTIFICATION

For the aleatoric uncertainty quantification experiment, we use the same hyperparameter and settings as Tagasovska *et al.* [6]. That is, we use a network with 1 ReLU activated hidden layer, with 64 hidden neurons and train it for 5000 epochs. We perform this for 20 seeds and for a learning rate $\eta \in \{10^{-2}, 10^{-3}, 10^{-4}\}$ and weight decay $\in \{0, 10^{-3}, 10^{-2}, 10^{-1}, 1\}$. For the input standard deviation, we made a single initial run with input variance $\sigma^2 \in \{10^{-8}, 10^{-7}, 10^{-6}, 10^{-5}, 10^{-4}, 10^{-3}, 10^{-2}, 10^{-1}, 10^0\}$ and then (for each data set) used 11 variances at a resolution of $10^{0.1}$ around the best initial variance.

### D.3    EPISTEMIC UNCERTAINTY FOR SELECTIVE PREDICTION ON OUT-OF-DISTRIBUTION DATA

Here we use the same network architecture as for the robustness experiments in the next subsection (A CNN composed of two ReLU-activated convolutional layers with a kernel size of 3, a stride of 2, and hidden sizes of `64` and `128`; followed by a convolutional layer mapping it to an output size of 10) and also train it for 100 epochs, however, at a learning rate of $10^{-3}$. For training with our uncertainty propagation, we use an input standard deviation of $\sigma = 0.1$. For training with uncertainty propagation via moment matching and the Dirichlet loss as in Gast *et al.* [11], we use an input standard deviation of $\sigma = 0.01$, which is what they use in their experiments and also performs best in our experiment.

### D.4    GAUSSIAN AND ADVERSARIAL NOISE ROBUSTNESS

Here, our CNN is composed of two ReLU-activated convolutional layers with a kernel size of 3, a stride of 2, and hidden sizes of `64` and `128`. After flattening, this is followed by a convolutional layer mapping it to an output size of 10. We have trained all models for 100 epochs with an Adam optimizer, a learning rate of $10^{-4}$, and a batch size of 128.

### D.5    PROPAGATING WITHOUT COVARIANCES

The architecture of the 5-layer ReLU-activated FCNN is **784−256−256−256−256−10**.

### D.6    EVALUATION METICS

#### D.6.1    COMPUTATION OF TOTAL VARIATION IN THE SIMULATION EXPERIMENTS

We compute the total variation using the following procedure: We sample $10^6$ samples from the continuous propagated distribution. We bin the distributions into $10 \times 10 \times 10$ bins (the output is 3-dimensional) and compute the total variation for the bins. Here, we have an average of 1000 samples per bin. For each setting, we compute this for 10 input points and on 10 models (trained with different seeds) each, so the results are averaged over 100 data point / model configurations. Between the different methods, we use the same seeds and oracles. To validate that this is reliable, we also tested propagating $10^7$ samples, which yielded very similar results, but just took longer to compute the oracle. We also validated the evaluation methods by using a higher bin resolution, which also yielded very similar results.

### D.6.2 COMPUTATION OF MPIW

Given a predicted mean and standard deviation, the Prediction Interval Width (for $95\%$) can be computed directly. In a normal distribution, $68.27\%$ of samples lie within $1$ standard deviation of the mean. For covering $95\%$ of a Gaussian distribution, we need to cover $\pm 1.96$ standard deviations. Therefore, the PIW is $3.92$ standard deviations. We computed the reported MPIW on the test data set. This setup follows the setup by Tagasovska et al. [6].

### D.7 MAXPOOL

For MaxPool, we apply Equation 1 such that $(\mu, \sigma) \mapsto (\max(\mu), \arg\max(\mu) \cdot \sigma)$ where $\arg\max$ yields a one-hot vector. The equation maxpools all inputs to one output; applying it to subsets of neurons correspondingly maxpools the respective subsets.

