# OpenReview forum: "Propagating Distributions through Neural Networks"
_ICLR.cc/2022/Conference — ICLR 2022 Submitted_

### Official Review · Reviewer_rdMQ · 2021-10-29

**Correctness:** 4
**Technical Novelty And Significance:** 3
**Empirical Novelty And Significance:** 3
**Recommendation:** 6
**Confidence:** 4

**Main Review:**

Strength:
1. The proposed method is very simple, and has theoretical justifications with respect to total variation. The method is shown to be applied to relatively larger neural network structures where some previous methods are intractable.

2. The improvements on the networks' robustness are interesting. This may bring more focus about studying the robustness and uncertainty of a network together.

Weakness:
1. The proposed method is only shown to have theoretical justifications for ReLU non-linearity. I think to extend it to more general non-linearities or give some more empirical analysis about the approximation, at least for the class of piece-wise linear functions, is important for this work. Also, it is not clear how to apply the method on max-pooling layers.

2. Even though the method is simple, and efficient to some extend, the running time is still quite significant for large Resnet structures (with covariance). This limits the applicability of the proposed method to more realistic settings and datasets.

3. The experimental settings are quite limited. Especially for the out-of-distribution data prediction, I think it's worth investigating more datasets, and different ways of introducing OOD data: for example, separating a dataset into different classes, or creating certain artifacts on a portion of the data.

4. Given the network structure that the method can apply, the network is a piecewise linear model overall. Particularly, the model is a linear model for every data point. What will be the performance of the method if we propagate the distribution through every local linear model?

**Summary Of The Paper:**

The paper proposes a simple method that propagates the uncertainty in the data through a neural network with ReLU non-linearity. The proposed method simply transforms the mean and covariance of the input Gaussian/Cauchy distribution through linear layers, and applies a local linear approximation for the ReLU activation. The proposed transformation is shown to be optimal under the total variation criterion. The paper applies the method to train neural networks with regression or classification targets, and show that the proposed method can well estimate the uncertainty of the network's output. Moreover, they also show some improvements in the robustness of the trained networks.

**Summary Of The Review:**

The main strength of the paper is that the proposed method is simple. The main weakness is that the method is only applicable to limited kinds of layers of neural networks, and the paper lacks more comprehensive experiments to demonstrate the method's performance.

---

> ### Author Response · Authors · 2021-11-18
> **Response (2/2)**
>
> > Given the network structure that the method can apply, the network is a piecewise linear model overall. Particularly, the model is a linear model for every data point. What will be the performance of the method if we propagate the distribution through every local linear model?
>
> If we understood the question correctly, we believe you are asking about a method that would propagate the distribution through every local linear model separately? [C] considered a related idea to compute the output distribution exactly: they divided the input space into linear regions and for each linear region (i.e., local linear model) propagated the respective section of the distribution. However, the computational complexity is exponential in the number of neurons, and the largest model that they could train had around 20-30 neurons. We briefly discussed [C] in Section 5.5 ([40] in the draft). Please let us know if we misunderstood your question.
>
>
> Please let us know whether you have further suggestions and questions.
> We are looking forward to hearing back from you.
>
> ---
>
> **[A]** Maddox, W.J., Izmailov, P., Garipov, T., Vetrov, D.P. and Wilson, A.G., 2019. A simple baseline for Bayesian uncertainty in deep learning.
>
> **[B]** Madry, A., Makelov, A., Schmidt, L., Tsipras, D. and Vladu, A., 2017. Towards deep learning models resistant to adversarial attacks.
>
> **[C]** R. Balestriero, S. Paris, and R. G. Baraniuk, “Analytical Probability Distributions and EM-Learning for Deep Generative Networks,” in the Proceedings of Neural Information Processing Systems (NIPS), 2020.

---

> ### Author Response · Authors · 2021-11-18
> **Response (1/2)**
>
> Thank you very much for your review.
> We thank you for your appreciation for the simplicity, theoretical justification, and tractability of our method.
>
> > The proposed method is only shown to have theoretical justifications for ReLU non-linearity. I think to extend it to more general non-linearities or give some more empirical analysis about the approximation, at least for the class of piece-wise linear functions, is important for this work.
>
> As for the theory, we have not been able to prove optimality for any of the other activations.
> We would like to add that for strictly monotonic activation functions (e.g., LeakyReLU, SoftPlus, Sigmoid) the linearization method is guaranteed to yield the exact probability density at the mean/median.
> This can be seen via the change of variables.
> If you think that this would be helpful, please let us know, and we will include this in the supplementary material.
>
> As for the empirical evaluation, we would like to point out that we have performed an empirical analysis for the LeakyReLU, SiLU, and GELU non-linearities.
> Also, we have added an experiment for the sigmoid function in the revision (Table 22).
>
> > Also, it is not clear how to apply the method on max-pooling layers.
>
> According to equation (1), we apply the max-pooling in the following way:
> $(\mu, \sigma) \mapsto (\max(\mu), \operatorname{argmax}(\mu) \cdot \sigma)$
> where $\operatorname{argmax}$ yields a one-hot vector.
> Here, we maxpool all inputs to one output; applying it to subsets of neurons would correspondingly maxpool the respective subsets.
> We have included this in the Supplementary Material D.7.
>
> > Even though the method is simple, and efficient to some extend, the running time is still quite significant for large Resnet structures (with covariance). This limits the applicability of the proposed method to more realistic settings and datasets.
>
> We would like to point out that none of the alternative methods can even be used to train a ResNet architecture as (i) DVIA would require unrealistic amounts of computing power and memory and (ii) methods without covariance propagation (i.e., moment matching and ours w/o cov.) do not allow training deep networks with tens of layers due to vanishing variances.
> The computational complexity of propagating distributions with covariances is $k+1$ times bigger than the vanilla model, i.e., 11-times for results in Tables 4 and 5.
> While we do observe a 25-fold increase, the complexity analysis suggests that the run-time of the method can be improved with a more efficient implementation (note that the baseline in this case is a standard implementation which is already highly optimized).
> We would also like to point out that methods for accessing uncertainty are generally a lot more computationally expensive, e.g., popular BNN/ensemble-based methods typically require many Monte Carlo samples to estimate the uncertainty (each sample requires a full forward pass; [A] uses 30 samples for CIFAR10).
> Achieving adversarial robustness is usually even more computationally demanding with other methods (a single iteration requires solving an inner maximization problem involving many backward passes, and the overall convergence is a lot slower [B]).
> While any of these methods limit the applicability, our method provides a fair trade-off.
>
> > The experimental settings are quite limited. Especially for the out-of-distribution data prediction, I think it's worth investigating more datasets, and different ways of introducing OOD data: for example, separating a dataset into different classes, or creating certain artifacts on a portion of the data.
>
> We agree that it would be interesting to extend the OOD experiment in these directions in future work, but would like to point out that we have done 5 sets of experiments and that we already had to defer many experimental results to the supplementary material.

---

### Official Review · Reviewer_cdSE · 2021-10-30

**Correctness:** 3
**Technical Novelty And Significance:** 3
**Empirical Novelty And Significance:** 2
**Recommendation:** 6
**Confidence:** 4

**Main Review:**

The paper presents a simple new technique to propagate distributions through neural networks. A nice feature of the method is that it can be readily used for pre-trained models. To justify the proposed method, a theoretical argument is provided that for the ReLU activation function a total variation distance is minimized. The paper is mostly clear and well written but lacks some details (see below).

- Regarding experiments showing the total variation metric (i.e., Table 1 and several tables in the supplementary):

How is the total variation between 10^6 (discrete) samples and the (continuous) output of probability propagation computed? I have another concern about the significance of the total variation distance. In particular, it is mentioned in the introduction that f(x + \epsilon) is a random variable where the variance is used to quantify its uncertainty. How well are the prediction uncertainties between the proposed method and sampling 10^6 models in these experiments aligned? Or can the total variation distance be used to give guarantees about the predicted uncertainty in these experiments?

- Regarding activation functions:

How well does the proposed method perform with other activation functions. The experiments in the main paper are only concerned with ReLU activations and the experiments in the supplementary material are concerned with ReLU-like activation functions (leaky ReLU, GELU, ...). Does the proposed method also produce reasonable results for saturating activation functions such as tanh or the logistic sigmoid?

- Failure modes:

The paper presents mostly results where the method succeeds although there are some obvious failure modes. For instance, given a Gaussian input with either a mean very close to zero or a very large variance, the approximation made by the proposed method is not very accurate. However, there are some experiments where the injected variances are rather large (e.g., Table 1 or Figure 3) but still the method does not seem to break down. How can you explain this?

Furthermore, I see another failure mode regarding the proposed objective for classification where the sum over all classes is computed. I do not see why this objective is enforcing the model to make a correct prediction. If P(Z_c > Z_e) is close to one for all classes except for a single class where P(Z_c > Z_e) < 0.5, then a wrong prediction will be made but still a high score is assigned by the proposed loss. If, on the other side, P(Z_c > Z_e) is slightly above 0.5 for all classes, then the correct class will be predicted but only a low score is assigned by the proposed loss. This might be more severe for classification tasks with a very large number of classes (e.g., ImageNet). In my opinion, it would make more sense to define the score by maximizing "min_e P(Z_c > Z_e)". This is similar to the concept of a probabilistic margin, e.g., see [1].

- Regarding presentation:

Please emphasize somewhere in the paper that the only point where stochasticity is injected is at the inputs (i.e., no weight uncertainty, no injected noise at hidden neurons, etc). Although I (now) see that this is clearly stated in the introduction, I would really appreciate if a formal definition of the considered setup is made at the beginning of Section 3. If space is an issue, I think Figure 1 is not that important and can be moved into the supplementary material.

- Regarding experiments:

  - How do you set the input variances in your experiments in Section 5.1? Is the input variance tuned so that the PICP is close to 95%? Is the input standard deviation in Section 5.2 always 0.1?
  - How is the prediction interval in Section 5.1 computed? Is the MPIW computed at the test data? In general, I would appreciate a clearer explanation of the used metrics and the experimental setup in Section 5.1.
  - In Section 5.2 it is stated that training is performed using both Pairwise Gaussian and Pairwise Cauchy. However, I do not see any results for training with Pairwise Cauchy.
  - What architecture is used in Section 5.2?
  - Results on Cifar-10: It would be interesting to see how well prediction uncertainties work for CNNs? For instance, a similar experiment as in Section 5.2 for out-of-distribution detection would be interesting (e.g., using samples from Cifar-100 for out-of-distribution data).
  - An evaluation of "in-between uncertainties" [2] would be interesting.


- Regarding computational complexity:

Section 5.5 states that the complexity for propagating distributions with and without covariances is linear and constant in the number of outputs, respectively. How can the complexity be linear if there are quadratically many entries in the covariance matrix? Similarly, how can the complexity be constant if we need to compute a variance entry for each output?

- Notation:

  - x used multiple times (e.g., (i) as input value of a neural network in the introduction and (ii) as parameter of the Cauchy distribution)
  - \sigma used multiple times (e.g., (i) as standard deviation of a Gaussian and (ii) as standard deviation of the input noise)

- References:
  - [1] Roth et al., Hybrid Generative-Discriminative Training of Gaussian Mixture Models, Pattern Recognition Letters
  - [2] Foong et al., 'In-Between' Uncertainty in Bayesian Neural Networks, ICML 2019 Workshop on Uncertainty and Robustness in Deep Learning

**Summary Of The Paper:**

The paper proposes a method to propagate uncertainties through neural networks by performing a first-order approximation at nonlinear activation functions. By injecting uncertainty into the inputs, prediction uncertainties are obtained. These prediction uncertainties are exploited at training time by using new uncertainty-aware loss functions, and at test-time to obtained prediction uncertainties, out-of-distribution detection, and adversarial robustness.

**Summary Of The Review:**

The proposed method is a simple and effective method to propagate distributions through a neural network. I am fine with the experimental results. However, I am not voting for accept right now since in my opinion there are some important details missing (as outlined in my review above) that are required to reproduce results and that would improve the quality of the paper. I am also missing a discussion of failure modes.

--- POST REBUTTAL ---

I have updated my rating from 5 to 6.

---

> ### Author Response · Authors · 2021-11-18
> **Response (4/4)**
>
> > Is the input standard deviation in Section 5.2 always 0.1?
>
> Yes, for our methods it is always 0.1. For moment matching, we use 0.01, as this performs best for moment matching and is also what was used by Gast et al. [11] in their experiments.
>
> > How is the prediction interval in Section 5.1 computed? Is the MPIW computed at the test data?
>
> Given a predicted mean and standard deviation, the Prediction Interval Width (for 95%) can be computed directly. In a normal distribution, 68.27% of samples lie within 1 standard deviation of the mean. For covering 95% of a Gaussian distribution, we need to cover +- 1.96 standard deviations. Therefore, the PIW is 3.92 standard deviations.
> We computed the reported MPIW on the test data set. This setup follows the setup by Tagasovska et al.
> We have added this to Supplementary Material D.
>
> > In Section 5.2 it is stated that training is performed using both Pairwise Gaussian and Pairwise Cauchy. However, I do not see any results for training with Pairwise Cauchy.
>
> Thank you for noticing this typo. We meant to include Cauchy in the next sentence "four" -> "five methods to compute certainty scores" but accidentally included it in the prior sentence. We fixed it.
>
> > What architecture is used in Section 5.2?
>
> A CNN composed of two ReLU-activated convolutional layers with a kernel size of 3, a stride of 2, and hidden sizes of 64 and 128. After flattening, this is followed by a convolutional layer, mapping it to an output size of 10.
> This is also described in Supplementary Material D.
>
> > "Results on Cifar-10: It would be interesting to see how well prediction uncertainties work for CNNs? For instance, a similar experiment as in Section 5.2 for out-of-distribution detection would be interesting (e.g., using samples from Cifar-100 for out-of-distribution data).
>
> Thank you for the suggestion. We agree that this could be an interesting experiment for future work focusing on out-of-distribution selective classification. In the present paper, our goal is to demonstrate a variety of applications of our method, rather than focusing on a specific one. We presented experiments demonstrating utility of our method for both aleatoric and epistemic uncertainty quantifications, and robustness to adversarial and Gaussian noises.
>
> > An evaluation of "in-between uncertainties" [2] would be interesting.
>
> We agree that this also is an interesting experiment for future work.
>
> > Section 5.5 states that the complexity for propagating distributions with and without covariances is linear and constant in the number of outputs, respectively. How can the complexity be linear if there are quadratically many entries in the covariance matrix? Similarly, how can the complexity be constant if we need to compute a variance entry for each output?
>
> We give the complexity as a relative factor in comparison to regular propagation through a neural network, as this is what matters most. Otherwise, this would become a quite involved complexity, as the complete architecture has to be known to compute the cost of a network. (E.g., for ResNet, there are many factors to consider.)
> Therefore, giving the relative complexity is more meaningful than giving the absolute complexity. We clarified this more explicitly in the revision.
>
>
> > x used multiple times (e.g., (i) as input value of a neural network in the introduction and (ii) as parameter of the Cauchy distribution) / \sigma used multiple times (e.g., (i) as standard deviation of a Gaussian and (ii) as standard deviation of the input noise)
>
> Thank you for noting this. We have revised the notation for Cauchy distribution location parameter. Regarding \sigma, we find it appropriate to use it in both cases, since it refers to a standard deviation of a Gaussian distribution. We included the information on what distribution each standard deviation belongs to.
>
> > I am not voting for accept right now since in my opinion there are some important details missing (as outlined in my review above) that are required to reproduce results and that would improve the quality of the paper.
>
> We have included the additional information you requested in the revision. Please let us know if there are any other important implementation details that are missing.
>
> Please let us know whether you have further suggestions and questions.
> We are looking forward to hearing back from you.
>
> ---
>
> **[A]** L. Devroye, A. Mehrabian, and T. Reddad, “The total variation distance between high-dimensional gaussians,” arXiv preprint arXiv:1810.08693, 2018.
>
> **[B]** Wang, Hao, Xingjian Shi, and Dit-Yan Yeung. “Natural-parameter networks: A class of probabilistic neural networks.” in Proc. of NeurIPS, 2016.
>
> **[C]** https://math.stackexchange.com/questions/207861/expected-value-of-applying-the-sigmoid-function-to-a-normal-distribution
>
> **[D]** https://en.wikipedia.org/wiki/Logit-normal_distribution#Moments
>
> **[E]** https://cran.r-project.org/web/packages/logitnorm/logitnorm.pdf

---

> > ### Comment · Reviewer_cdSE · 2021-11-29
> > **Response**
> >
> > I thank the authors for their detailed reply. Most of my questions have been answered and I am changing my rating from 5 to 6.

---

> ### Author Response · Authors · 2021-11-18
> **Response (3/4)**
>
> > Furthermore, I see another failure mode regarding the proposed objective for classification where the sum over all classes is computed. I do not see why this objective is enforcing the model to make a correct prediction. If P(Z_c > Z_e) is close to one for all classes except for a single class where P(Z_c > Z_e) < 0.5, then a wrong prediction will be made but still a high score is assigned by the proposed loss. If, on the other side, P(Z_c > Z_e) is slightly above 0.5 for all classes, then the correct class will be predicted but only a low score is assigned by the proposed loss. This might be more severe for classification tasks with a very large number of classes (e.g., ImageNet). In my opinion, it would make more sense to define the score by maximizing "min_e P(Z_c > Z_e)". This is similar to the concept of a probabilistic margin, e.g., see [1].
>
> This is an interesting thought. However, this does not only apply to the proposed loss but instead also to the conventional softmax cross entropy loss. In the following, we show such an example for softmax cross entropy where a smaller loss is given to the wrong prediction. Let the true class be the one with index 0. Consider the following neural network outputs (i.e., pre-softmax values):
>
> * correct prediction [0.1, 0, 0, 0, 0] -> softmax CE loss = 1.5303
>
> * incorrect prediction: [0, 0.1, -1, -1, -1] -> softmax CE loss = 1.1659
>
> While such effect could theoretically cause a worse model to be selected if model selection is done based on the loss, this is usually not problematic in learning (e.g., ImageNet w/ softmax CE loss).
> Also, because of the sum (/mean), the pairwise losses could be considered to be independent losses and thus, even if all except for one loss component are saturated, the one (last) loss is still being optimized and is not impacted by this.
> In an analogy, if we have multiple summed loss components in a batch and most of the individual loss components are small, this does not hinder the remaining large loss components from being effectively optimized.
>
> The additional application of a probabilistic margin is an interesting idea for future work.
>
> > Regarding presentation: Please emphasize somewhere in the paper that the only point where stochasticity is injected is at the inputs (i.e., no weight uncertainty, no injected noise at hidden neurons, etc). Although I (now) see that this is clearly stated in the introduction, I would really appreciate if a formal definition of the considered setup is made at the beginning of Section 3. If space is an issue, I think Figure 1 is not that important and can be moved into the supplementary material.
>
> Thank you for the suggestion. We have added a comment at the beginning of Section 3 to clarify this. We will add a more formal statement in subsequent revisions.
>
> > How do you set the input variances in your experiments in Section 5.1? Is the input variance tuned so that the PICP is close to 95%?
>
> Yes, the input variance is selected so that the PICP is close to 95% on the validation set. The displayed results are for the test set. This procedure was also applied to all baselines.

---

> ### Author Response · Authors · 2021-11-18
> **Response (2/4)**
>
> > How well does the proposed method perform with other activation functions. The experiments in the main paper are only concerned with ReLU activations and the experiments in the supplementary material are concerned with ReLU-like activation functions (leaky ReLU, GELU, ...). Does the proposed method also produce reasonable results for saturating activation functions such as tanh or the logistic sigmoid?
>
> To answer this question, we performed the experimental setting from Table 1 for the logistic sigmoid function.
> For the baseline of this, we had to implement moment matching for logistic sigmoid.
> As it does not have a closed form solution and only approximations are known in the literature, we implemented a method for computing it via numerical integration (using $10\,000$ steps to be on the safe side). (In case you are interested, [B] and [C] provide approximations, [D] also discusses that there are no analytical solutions.)
> DVIA is not applicable, as it is only for ReLU and Heaviside functions.
>
> The results below show that our method with covariances yields the best approximations for small input standard deviations, and for large standard deviations is equivalent to without covariances.
> Moment matching performs similar to ours w/o cov. for small input stds. and underperforms it for larger stds.
> Overall, the results show that propagating Gaussian distributions through logistic sigmoids works reasonably well.
>
> | $\sigma$ | ours (w/ cov.) | ours (w/o cov.)  | Moment Matching   |
> |------|-------|-------|-------|
> |  0.1 | 0.9890 | 0.7809 | 0.7793 |
> |    1 | 0.9562 | 0.7880 | 0.7912 |
> |   10 | 0.8647 | 0.8656 | 0.7674 |
> |  100 | 0.8443 | 0.8442 | 0.8027 |
> | 1000 | 0.8440 | 0.8440 | 0.8070 |
>
> We have included the results in the supplementary material of the paper, and we will also include our implementation of sigmoid-moment matching when we release the PyTorch source code of our work.
>
> > For instance, given a Gaussian input with either a mean very close to zero or a very large variance, the approximation made by the proposed method is not very accurate. However, there are some experiments where the injected variances are rather large (e.g., Table 1 or Figure 3) but still the method does not seem to break down. How can you explain this?
> Of course, this always depends on the notion of breaking down.
>
> In Table 1, we have a 68% intersection of probability mass between the oracle and our approximation.
> In some sense, this could be considered "breaking down" (in comparison to 98% for smaller input standard deviations), but indeed in comparison to any of the methods without covariances this is a very good performance.
> The methods without covariances (without cov. (ours) and moment matching) could be considered "breaking down" in all cases because the covariances play a big role (in the ReLU case) and ignoring them leads to underestimating the variances.
> The first table of our response as well as Figure 8 in the draft demonstrate this underestimation when ignoring covariances.
> This leads to a large TV and a small probability mass intersection, as the distribution is vastly differently shaped (having way too small variances).
> The importance of covariances makes it therefore harder to compare methods with and without covariances, as those with covariances have a significant advantage.
> Please let us know whether this clarifies your concern.
>
> As for Table 3, whether a method breaks down mostly depends on whether the loss and the propagation are still numerically stable for a given magnitude of input uncertainty.
> Notice how the moment matching method (red) is partially covered by the black softmax CE line and therefore also performs well over many orders of input std. magnitude. (we will fix this, by drawing the red line on top of the black line).
> That the methods work for a large range of input standard deviations is primarily because the training of neural networks are flexible and can account for this.
> Also, the pairwise loss works even when the probabilities are close to 0 or 1.
>
> If you think that such a discussion would be helpful, we will include this in the supplementary material. Please inform us, if you would like to see that.

---

> ### Author Response · Authors · 2021-11-18
> **Response (1/4)**
>
> Thank you very much for your extensive and thorough review.
> In our answer, we also include some new experiments providing empirical evidence to answer your questions.
>
>
> > How is the total variation between 10^6 (discrete) samples and the (continuous) output of probability propagation computed?
>
> The idea is to represent a continuous distribution as a discrete one by sampling many samples from it. Specifically, we compute the total variation using the following procedure: We sample 10^6 samples from the continuous propagated distribution. We bin the distributions into $10\times 10\times 10$ bins (the output is 3-dimensional) and compute the total variation for the bins. Here, we have an average of $1000$ samples per bin.
> For each setting, we compute this for 10 input points and on 10 models (trained with different seeds) each, so the results are averaged over 100 data point / model configurations. Between the different methods, we use the same seeds and oracles.
> To validate that this is reliable, we also tested propagating 10^7 samples, which yielded very similar results, but just took longer to compute the oracle.
> We also validated the evaluation methods by using a higher bin resolution, which also yielded very similar results.
>
> We have included this description in the supplementary material.
>
> > In particular, it is mentioned in the introduction that f(x + \epsilon) is a random variable where the variance is used to quantify its uncertainty. How well are the prediction uncertainties between the proposed method and sampling 10^6 models in these experiments aligned? Or can the total variation distance be used to give guarantees about the predicted uncertainty in these experiments?
>
> Total variation is a proper metric on probability distributions, i.e., TV distance of 0 implies that all moments (including variance) are the same. For Gaussians [A] show that the total variation bounds the maximum difference between the variances.
> To investigate the ratio between the predicted output standard deviations and the true output standard deviations (from the sampling oracle), we reproduced the experiment but instead of TV, we computed the average ratio between the standard deviations in the following table:
>
> | $\sigma$ | ours (w/ cov.)  | DVIA  | ours (w/o cov.)  | Moment Matching   |
> |------|-------|-------|-------|-------|
> |  0.1 | 0.9933,1.0043,0.9954 | 0.9904,0.9917,0.9906 | 0.0126,0.0180,0.0112 | 0.0255,0.0373,0.0211 |
> |    1 | 0.9248,1.0184,0.9598 | 0.8462,0.9256,0.8739 | 0.0103,0.0170,0.0100 | 0.0101,0.0166,0.0097 |
> |   10 | 1.9816,2.2526,2.0629 | 0.6035,0.6664,0.6172 | 0.0207,0.0359,0.0205 | 0.0166,0.0269,0.0157  |
> |  100 | 2.5404,2.8161,2.6217 | 0.3721,0.4194,0.3849 | 0.0264,0.0450,0.0260 | 0.0186,0.0296,0.0174  |
> | 1000 | 2.6095,2.8857,2.6906 | 0.3375,0.3839,0.3503 | 0.0271,0.0462,0.0267 | 0.0188,0.0298,0.0175  |
>
> Here, a value of 1 is optimal.
> Each column includes 3 values, which correspond to the three output dimensions of the Iris model.
> We find that both our method w/ cov. as well as DVIA achieve a good accuracy in this setting, while the methods which do not consider covariances (ours (w/o cov.) and Moment Matching) underestimate the output standard deviations by a large factor.
> For small input standard deviations, our method w/ cov. as well as DVIA perform better.
> For large input standard deviations, our method tends to overestimate the output standard deviation, while DVIA underestimates the standard deviation.
> The ratios for our method and DVIA (while going into opposite directions) are similar, e.g., 2.6095 for our method and 1/0.3375=2.9629 for DVIA.
>
> We have added as Table 23.

---

### Official Review · Reviewer_H6dp · 2021-10-31

**Correctness:** 4
**Technical Novelty And Significance:** 2
**Empirical Novelty And Significance:** 2
**Recommendation:** 6
**Confidence:** 2

**Main Review:**

Strengths:
- Interesting theoretical result showing that the local linearization is in fact an optimal approximation in terms of total variation for ReLU networks
- Nice application of the method to study uncertainty quantification in neural networks

Weaknesses:
- Only consider the case where the activation function is ReLU (going beyond ReLU, can the linearization method still give reasonable approximations, under some conditions? Would be nice to have some theoretical results in this direction)

Comments/questions:
- What is the role of depth in the uncertainty quantification and improvement in robustness to random/adversarial perturbations?
- Can one formulate new loss function for learning with other non-Gaussian distributions (other than only Cauchy)?
- On page 3 (beginning of Section 3.1), is there a typo? There, A is a m by n matrix and b is a 1 by n vector?

**Summary Of The Paper:**

This paper studies the problem of propagating probability distributions through neural networks and applies the results to quantify prediction uncertainties. It proposes a local linearization method to approximate a distribution transformed by a ReLU network, as well as new loss function for learning with distribution-valued inputs. It also provides empirical results, showing that the method can quantify two kinds of uncertainty (aleatoric and epistemic) in classification and regression tasks, and training with the new loss function can improve robustness to random and adversarial perturbations.

**Summary Of The Review:**

Overall, the strengths of the paper outweigh the weaknesses, although going beyond the ReLU activation (see Weaknesses above) could significantly improve the quality of the paper. Therefore, I am inclined to accept the paper.

---

> ### Author Response · Authors · 2021-11-18
> **Response**
>
> Thank you very much for your review.
>
> > Only consider the case where the activation function is ReLU (going beyond ReLU, can the linearization method still give reasonable approximations, under some conditions? Would be nice to have some theoretical results in this direction)
>
> We would like to start by pointing out that we do empirically use other activation functions and obtain reasonable approximations.
> As for the theory, we have not been able to prove optimality for any of the other activations.
> We would like to add that for strictly monotonic activation functions (e.g., LeakyReLU, SoftPlus, Sigmoid) the linearization method is guaranteed to yield the exact probability density at the mean/median.
> This can be seen via the change of variables.
> If you think that this would be helpful, please let us know, and we will include this in the supplementary material.
>
> > What is the role of depth in the uncertainty quantification and improvement in robustness to random/adversarial perturbations?
>
> We are assuming depth refers to the number of layers.
> In Supplementary Material C.1, we empirically test how well distributions are propagated through neural networks with 1, 2, 4, and 6 hidden layers.
> While methods without covariances (i.e., ours w/o cov. and moment matching) suffer more from deeper models, the methods with covariances (i.e., ours w/ cov. and DVIA) perform well even for more layers.
> Comparing our method to DVIA, DVIA performs a little better for small numbers of layers (1 / 2 hidden layers), while for more layers (4 / 6 hidden layers), our propagation performs better.
> Our method is therefore more stable for varying numbers of layer / deeper networks.
> Regarding the robustness experiments on the real data - we used the standard architectures for the respective problems and did not study the effect of depth.
>
> > Can one formulate new loss function for learning with other non-Gaussian distributions (other than only Cauchy)?
>
> Yes, for example for the Laplace distribution, we could also evaluate P(X>Y) via the CDF. This also applies to other distributions.
>
> > On page 3 (beginning of Section 3.1), is there a typo? There, A is a m by n matrix and b is a 1 by n vector?
>
> Thanks for noticing the typo, b is a 1 by **m** vector.
>
> Please let us know whether you have further suggestions and questions.
> We are looking forward to hearing back from you.

---

### Official Review · Reviewer_bo8J · 2021-11-02

**Correctness:** 4
**Technical Novelty And Significance:** 2
**Empirical Novelty And Significance:** 2
**Recommendation:** 3
**Confidence:** 4

**Main Review:**

The research problem of improving uncertainty estimates is very interesting and useful. However, the idea of propagating probability distribution through neural networks is not new. Specifically, the introduction section is a bit inaccurate and misleading. The authors claim that they focus on obtaining the propagated distribution in analytical form, or closed form, and that prior work [17] focuses only on Gaussian distributions (note that [17] is very early work from 1999). This is not true. For example, natural-parameter networks, or NPN [23] as cited in the paper, as well as its variants [12,26], considers exactly the same problem, and already handles distributions beyond Gaussian distributions; in fact their methods could handle any exponential-family distributions (including Gaussian ones) as well as various activation functions beyond ReLU.

This brings me to another of my concerns, which is the lack of proper baselines. Given the similarity the authors should at least include baselines from [23, 12, 26]. That would make the paper more convincing and maybe highlight what local linearization is a better solution. Currently, it seems [27] is the only modern baseline method (and note that [27] did not establish its improvement over methods such as [23, 12, 26]).

In terms of technical novelty, since the results for linear transformations are well known, the main novelty rests on the nonlinear part, i.e., the use of local linearization. Here the authors are able to prove that in terms of TV, local linearization is optimal in the Gaussian/ReLU case and Cauchy/ReLU case. In the introduction, the author claim that they are able to be ‘more general’ than some previous methods that can only handle Gaussian distributions. It is a bit disappointing to see that the proposed version can only handle Gaussian and Cauchy distributions and it has to be ReLU activation.

In Section 4.2, the authors mentioned that they can compute the exact probability, which is interesting. The trick is to change the softmax into a pariwise Gaussian comparison. However, it seems a lot of details are missing.

Table 4 is a bit confusing. It shows that the proposed method actually underperforms the vanilla softmax CE loss. It is also unclear what is the setting for Section 5.4. For example, what is the input variance for ‘Pairwise Gaussian (PG)’, does PG produce better uncertainty estimates, etc.

Minor:
Grammar issue: Page 4: Note that when we … we refer to Dirac’s delta distribution.




**Summary Of The Paper:**

This paper proposed an approach to propagate probability distribution through neural networks. In particular, the authors use local linearization to handle nonlinearity and show its optimality in terms of total variation for ReLU. Empirically they show that the proposed method can provide calibrated confidence intervals for regression problems and improve detection of OOD data in classification problems compared to some baselines.

**Summary Of The Review:**

Overall, the paper proposed a method for propagating uncertainty through neural network that could potentially be useful. However, it seems some important baselines are missing, the introduction is somehow misleading, the proposed method only works in very limited cases, i.e., Gaussian+ReLU and Cauchy+ReLU (while theoretical results on other activation such as sigmoid/tanh/other ReLU variants are not provided).

---

> ### Author Response · Authors · 2021-11-18
> **Response**
>
> Thank you very much for your review.
>
> We have revised the introduction to clarify that there are prior extensions that can also handle non-Gaussian distributions. However, none of the prior works based on the moment matching idea are applicable to Cauchy distribution due to its moments being undefined/infinite.
>
>
> > This brings me to another of my concerns, which is the lack of proper baselines. Given the similarity the authors should at least include baselines from [23, 12, 26]. That would make the paper more convincing and maybe highlight what local linearization is a better solution. Currently, it seems [27] is the only modern baseline method (and note that [27] did not establish its improvement over methods such as [23, 12, 26]).
>
> In our experiments, we compare to moment matching, which is a general baseline that is equivalent to many of the methods that you mentioned. For example, [A] shows how to apply moment matching to ReLU and Leaky ReLU. [B] generalizes moment matching to exponential family distributions. In our experiments with Gaussian input distributions and various activation functions (Table 1 and Tables 6-13), [A] and [B] and "moment matching" are equivalent.
>
> > In the introduction, the author claim that they are able to be ‘more general’ than some previous methods that can only handle Gaussian distributions. It is a bit disappointing to see that the proposed version can only handle Gaussian and Cauchy distributions, and it has to be ReLU activation.
>
> We would like to clarify that only our theoretical results are limited to ReLU. In our experiments and simulations, we do also use other activation functions such as LeakyReLU, SiLU, GELU, MaxPool, and (since the revision) Logistic Sigmoid.
> There are also no particular reasons against applying equation (1) or (2) with other activation functions, as long as they are differentiable or at least continuous.
> We would like to add that our proof for the Gaussian case applies to most other symmetric distributions, e.g., the Cauchy (as shown in the paper), the Laplace, the Logistic, or the Holtsmark distribution.
>
> > In Section 4.2, the authors mentioned that they can compute the exact probability, which is interesting. The trick is to change the softmax into a pariwise Gaussian comparison. However, it seems a lot of details are missing.
>
> Could you please elaborate on which details are missing? Are you referring to a loss function? That would be minus equation (11). Are you referring to a derivation of the probability P(X>Y)?
> We are happy to answer any specific questions about this section and include the respective information.
> If you would like to see a derivation, we could include it in the supplementary material.
>
> > Table 4 is a bit confusing. It shows that the proposed method actually underperforms the vanilla softmax CE loss. It is also unclear what is the setting for Section 5.4. For example, what is the input variance for ‘Pairwise Gaussian (PG)’, does PG produce better uncertainty estimates, etc.
>
> The purpose of Table 4 is to demonstrate that our method can handle large ResNets of 18 and 34 layers.
> This contrasts moment matching, with which it was not possible to train such deep architectures because training diverged, as well as other methods, with which it would be infeasible to train in such settings because of computational limitations.
> Note that other methods propagating covariances do have a complexity overhead factor in the number of neurons (e.g., 12500 for our small CNN and much larger for ResNets), while our method only has a complexity factor in the number of outputs (e.g., 10).
> This makes it impossible to train such large models due to memory and time constraints. Given infinite memory, comparable methods are at least 1000 times slower for a small CNN and even more for ResNets.
> The focus here is on demonstrating the scalability, and we already have aleatoric uncertainty, an epistemic uncertainty, and multiple robustness experiments. Thus, we do not include this kind of analysis in the scalability experiments.
>
> Thank you for pointing out the typo.
>
> > the proposed method only works in very limited cases, i.e., Gaussian+ReLU and Cauchy+ReLU
>
> We would like to clarify that the proposed method *works* also in other cases, as demonstrated in the paper.
> The limitation to Gaussian+ReLU and Cauchy+ReLU only applies to the theoretical result.
>
> Please let us know whether you have further suggestions and questions.
> We are looking forward to hearing back from you.
>
> ---
>
> **[A]** A. Shekhovtsov and B. Flach, “Feed-forward propagation in probabilistic neural networks with categorical and max layers,” in Proc. Int. Conf. on Learning Representations (ICLR 2019, New Orleans, LA), 2019.
>
> **[B]** Wang, Hao, Xingjian Shi, and Dit-Yan Yeung. “Natural-parameter networks: A class of probabilistic neural networks.” in Advances in Neural Information Processing Systems 29 (2016): 118-126.

---

### Author Response · Authors · 2021-11-18
**Revision**

We thank all reviewers for taking the time to review and providing interesting and extensive reviews.
We made a revision to our paper, with the following main changes:

* We clarified that moment matching can also be applied to other distributions, which have moments.
* We added a new experiment for the logistic sigmoid function in Table 22 in the Supplementary Material C.
* We added an additional comparison metric for the experiment in Table 1, where we investigate the average ratio between output standard deviations produced by the oracle and each respective method. This is in Table 23.
* We added the requested additional implementation details to Supplementary Material D.
* We added additional clarifications as suggested.
* We fixed typos, notation, etc.

We answer all your questions below in the respective responses.
Please let us know whether you have further suggestions and questions.

We are looking forward to hearing back from you.

---

### Decision · Program_Chairs · 2022-01-20

**Decision:**

Reject

**Comment:**

The paper discusses propagating input uncertainty through non-linear layers by a simple local linearization approach. This is a straightforward idea and the authors explain how this is an optimal approximation of the propagated distribution for Total Variation and reLU non-linearity (for a single layer). This is an interesting (if quite limited) theoretical result. What is not clearly stated is that this result only holds for a single layer. It does not mean that the local approach is the best way to approximate (in the total variation sense) a distribution passed through multiple reLU layers.

By repeating the procedure, the authors are able to define closed form objective functions for noise-robust training of deep networks.

The reviewers found this an interesting paper and there was a good effort by the authors to improve the results. However, technical innovation is modest and reviewer doubts still remain. For that reason, clarity of presentation is critically important. The overall numerical score isn't convincing and with one reviewer remaining very unconvinced.

I agree with the reviewers that the technical contribution is quite limited and I would argue is not particularly well explained. For example, a simple alternative would be to use a "global" linearization in which one can consider the network function $f$ as a whole, and then simply linearize this (rather than linearizing each layer). Indeed, the way that the paper is written, this would be a natural interpretation since $f$ is defined in the introduction as the network function, but is used later differently (eg section 3.2) as the transfer function. The approach is to recursively compute a new mean and covariance for each layer, propagating these through the network (similar to moment matching approaches). It would have helped if the authors had made pseudocode for their approach. It would be natural and interesting to compare to the simple global linearization approach (which is computationally faster).

The presentation of results and experiments could be improved. For example, in figure 3, it is not clear (nor is it explained in the text) what the definition of "robust accuracy" is.

Given the modest technical innovation, I also feel that the clarity of presentation isn't yet at the level that would merit acceptance. The paper would also benefit from some deeper insight into why the approach might perform better than other approaches (such as local moment matching) at the network level (rather than a single layer).